# Spatial profiling of microbial communities by sequential FISH with error-robust encoding

Zhaohui Cao [1,2,3], Wenlong Zuo [1,3], Lanxiang Wang[1], Junyu Chen[1], Zepeng Qu[1], Fan Jin [1,2] & Lei Dai [1,2] ✉

Spatial analysis of microbiomes at single cell resolution with high multiplexity and accuracy has remained challenging. Here we present spatial profiling of a microbiome using sequential error-robust fluorescence in situ hybridization (SEER-FISH), a highly multiplexed and accurate imaging method that allows mapping of microbial communities at micron-scale. We show that multiplexity of RNA profiling in microbiomes can be increased significantly by sequential rounds of probe hybridization and dissociation. Combined with error-correction strategies, we demonstrate that SEER-FISH enables accurate taxonomic identification in complex microbial communities. Using microbial communities composed of diverse bacterial taxa isolated from plant rhizospheres, we apply SEER-FISH to quantify the abundance of each taxon and map microbial biogeography on roots. At micron-scale, we identify clustering of microbial cells from multiple species on the rhizoplane. Under treatment of plant metabolites, we find spatial re-organization of microbial colonization along the root and alterations in spatial association among microbial taxa. Taken together, SEER-FISH provides a useful method for profiling the spatial ecology of complex microbial communities in situ.

Spatial structure of microbial communities has been observed across different habitats, ranging from marine biofilms[1], human gastrointestinal tracts and oral cavities[2,3], to plant phyllosphere and rhizosphere[4,5]. For example, microbial localization and density vary widely in animal guts (along both longitudinal and transverse axes), as well as in plant compartments, due to spatial heterogeneity in chemical and oxygen gradients, nutrient availability, and immune effectors[6,7]. Despite advances in high-throughput sequencing technologies, understanding of the spatial organization of complex microbial communities is still limited[8]. Development of highly multiplexed methods for system-level, spatially-resolved profiling of microbial communities is crucial to elucidate principles governing the assembly and functions of microbiomes, as well as their interactions with the environment and hosts[9–12].

Fluorescence in situ hybridization (FISH) with probes targeting ribosomal RNA (rRNA) has been widely used to identify specific microbial taxa and allows for in situ spatial analysis of microbiomes at single cell resolution[13–19]. One challenge of this spatial profiling is the huge phylogenetic and functional diversity of free-living and host-associated microbial communities. Hundreds to thousands of microbial species reside in soil, plant rhizospheres[20], and mammalian guts[21]. In situ profiling of meta-transcriptomes is even more challenging, as the estimated complexity of metagenomes (e.g., over 20 million genes in the human gut microbiome[22]) far exceeds the complexity of host genomes. Multiple methods have been developed to increase multiplexity in spatial mapping of microbiomes[23–28]. Combinatorial labeling and spectral imaging in CLASI-FISH allowed the number of imaged species to exceed the number of fluorophores[24,25]. The two-step

[1]CAS Key Laboratory of Quantitative Engineering Biology, Shenzhen Institute of Synthetic Biology, Shenzhen Institute of Advanced Technology, Chinese Academy of Sciences, Shenzhen 518055, China. [2]University of Chinese Academy of Sciences, Beijing 100049, China. [3]These authors contributed equally: Zhaohui Cao, Wenlong Zuo. ✉e-mail: lei.dai@siat.ac.cn

hybridization scheme and advanced spectral unmixing in HiPR-FISH further increased multiplexity[28]. However, the multiplexity of currently available imaging methods for microbiome samples is still inherently limited by the number of fluorophores, i.e., up to $2^F-1$ targets with F fluorophores. Methods based on sequential FISH can significantly increase multiplexity[29–33] and are very much needed for spatially resolved measurements of microbial communities.

Another challenge of applying FISH methods to profile complex microbial communities is the accuracy of target identification. The accuracy of taxonomic identification with FISH depends on the specificity of probe targeting, yet rRNA sequences of species closely related phylogenetically are highly similar. In a diverse microbial community, probes of nonspecific binding cannot be ignored; therefore, it is difficult to design perfectly selective probes at the species or sub-species level[12,34], and to achieve highly accurate target identification with multiplexed FISH methods. Error-correction strategy that can improve accuracy for target identification in the context of microbiome imaging is highly desirable, but remains largely unexplored.

Here we introduce sequential error-robust fluorescence in situ hybridization (SEER-FISH), a highly multiplexed and accurate microbiome imaging approach allowing spatial mapping of microbial communities at single cell resolution. We developed an experimental method that allows for multiple rounds of FISH imaging of microbial samples. The exponential combination of fluorophore numbers (F) and hybridization rounds (N) leads to an unparalleled increase in multiplexity ($F^N$) for labeling microbiome samples. By incorporating

error-robust encoding schemes, we showed that SEER-FISH could tolerate probe non-specificity to achieve high precision and recall in taxonomic identification of microbial communities. Finally, we applied SEER-FISH in imaging *Arabidopsis thaliana* roots to unravel the micron-scale biogeography of microbial communities colonizing the rhizoplane.

## Results
### Superior multiplexity of SEER-FISH in spatial profiling of microbiome
Fluorescent in situ hybridization (FISH) with probes targeting ribosomal RNA has been widely used for identification of specific microbial taxa. By encoding each target taxon with a unique barcode through N rounds of FISH imaging, SEER-FISH provides a scalable coding capacity of $F^N$ (F-color, N-round) (Fig. 1a). To sequentially label the target taxon with an N-bit barcode, we developed a protocol that allowed for iterative labeling of microbial rRNAs with rapid probe hybridization and dissociation (Supplementary Fig. 1, see Multi-round FISH imaging in Methods). In each round, probes of F colors were hybridized to targeted rRNAs, and the sample was imaged and then treated with dissociation buffer to remove the hybridized probes[31,35]. After experiments were completed, the images were aligned to eliminate the shift in position during multiple rounds of imaging. The boundaries of bacterial cells were segmented using the watershed algorithm, and the fluorescence intensity of bacterial cells in each round of imaging was determined to identify their corresponding barcodes (Supplementary Fig. 2).

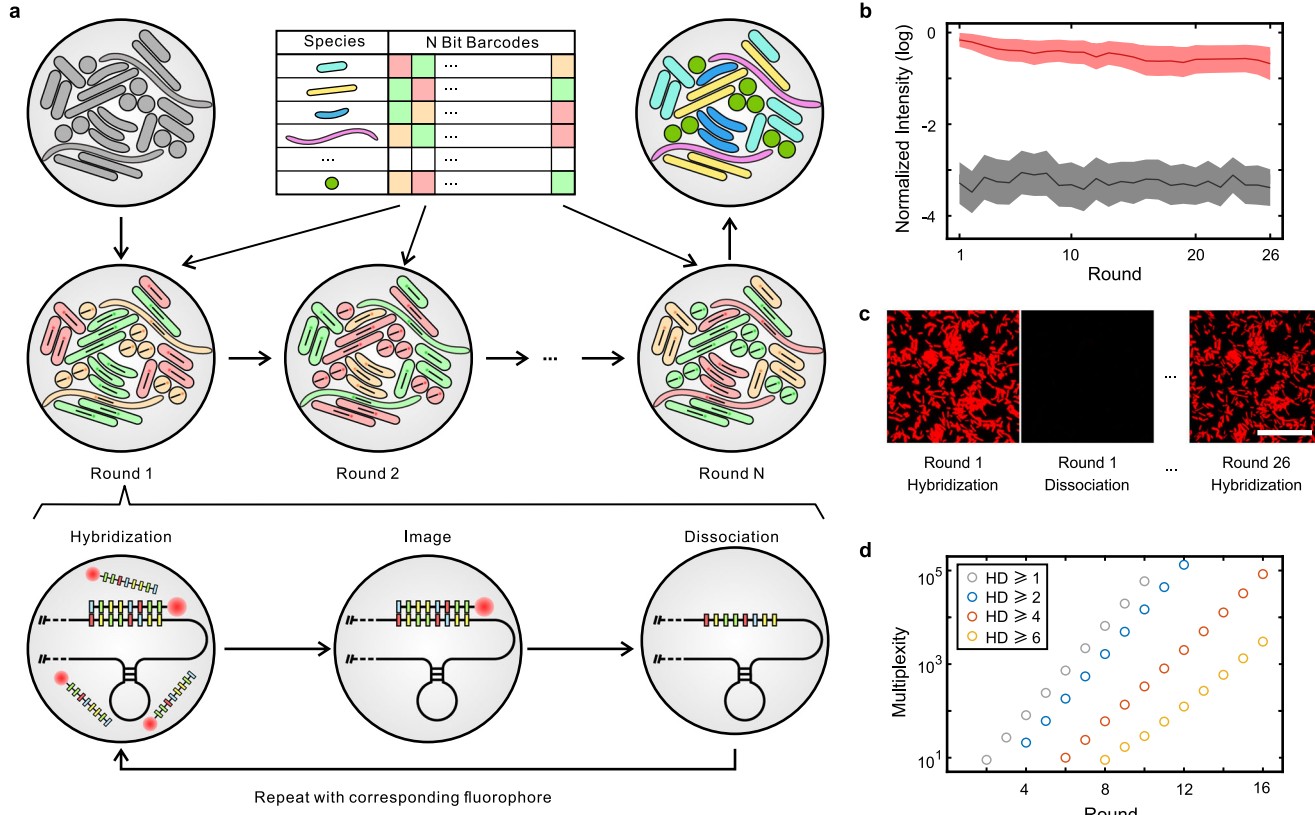

**Fig. 1 | SEER-FISH allows superior multiplexity in spatial profiling of microbiomes. a** Design of SEER-FISH. Each bacterial taxon is encoded by an F-color N-bit barcode. The spatial distribution of the microbial community can be obtained through R rounds of FISH. Each round of SEER-FISH includes probe hybridization, imaging, and probe dissociation (see Multi-round FISH imaging in Methods, Supplementary Fig. 1a). **b**, **c** Fluorescence intensity over 26 rounds of SEER-FISH. Lines indicate the mean fluorescence intensity (log-transformed and normalized by the

maximum pixel value of CCD) of bacterial cells (*n* = 2257) after hybridization (red line) and dissociation (black line), respectively. The shadow of each line indicates the standard deviation. Fluorescence intensity at the 1st and 26th rounds of imaging is shown in panel **c**. Scale bar, 25 μm. **d** The multiplexity of SEER-FISH increases exponentially with the number of rounds. The colors of the circles indicate the minimal Hamming distance (HD) between barcodes. All codebooks are generated with three colors (*F* = 3). Source data are provided as a Source Data file.

To evaluate the feasibility of our experimental method, we performed multiple rounds of FISH imaging on a mixture of bacterial species (SynCom12) with the universal probe EUB338 (Fig. 1b, c). The hybridized probes were efficiently removed by dissociation buffer, leading to an -1000-fold decrease in the fluorescence intensity. The dissociation step in our protocol had little effect on subsequent rounds of hybridization (Fig. 1b). In contrast, re-hybridization of probes after photobleaching was inefficient and did not allow for multiple rounds of imaging (Supplementary Fig. 3). After 26 rounds of probe hybridization and dissociation, we found that the mean fluorescence intensity of bacterial cells was still significantly higher than the background (Fig. 1c). Only a small fraction of bacterial cells (less than 3%) were lost due to the decrease in fluorescence intensity or the shift in their location.

Thus, for the first time as we know of, we developed an efficient method to label microbial rRNAs using sequential hybridization and dissociation of probes. Our protocol supports sequential FISH on microbial samples for more than 25 rounds, and each round of imaging takes only -15–30 min (Supplementary Fig. 1). In contrast, other methods of one-round bacterial FISH often require more than 2 h[15-17,24-28]. The coding capacity of SEER-FISH, similar to sequential FISH studies in the context of single cell transcriptomics[29-31], possesses great scalability through increasing the rounds of imaging ($F^N$) (Fig. 1d). A detailed comparison of existing methods for imaging microbiome samples, including CLASI-FISH[25] and HiPR-FISH[28], can be found in Supplementary Table 1. The sequential labeling of microbiome and error correction strategies developed by SEER-FISH can be complemented by the two-step probe design in HiPR-FISH[28] to take advantage of these complementary approaches in spatial mapping of microbiomes.

## Error-robust encoding enables high accuracy in taxonomic identification

One challenge for sequential FISH is that detection errors would increase with rounds of hybridization. Analogous to the strategy previously proposed in the context of labeling mRNAs in mammalian cells[32,33], we designed an error-robust encoding scheme that used a subset of the $F^N$ barcodes with specified minimal Hamming distance (HD) (Figs. 2a, 1d, see Codebook generation in Methods). For example, any two barcodes from a codebook with a minimal HD of 4 (HD4) differ by at least 4 bits. Therefore, we can correct 1-bit and some 2-bit detection errors by identifying the observed barcode compared to its nearest valid barcodes (Fig. 2b). Each bacterial taxon is labeled with a specific fluorescence color in each round, e.g., FAM/Cy3/Cy5($F = 1/2/3$), which is decoded by comparing brightness across different fluorescence channels (see Barcode identification in Methods).

To evaluate the feasibility of error-robust encoding schemes, we performed SEER-FISH on pure cultures of 12 bacterial species (Supplementary Table 2) with a set of R8HD4 barcodes (rounds = 8, minimal HD = 4, Fig. 2a). The design and selection of probes specifically targeting 16 S or 23 S rRNA of the corresponding bacterial species are based on stringent criteria that take into account sequence mismatch to non-target taxa[36] and predicted hybridization efficiency to the targeted taxa[37] (Supplementary Fig. 4). Each bacterial species was separately coated onto a coverslip, hybridized with probes according to the codebook and imaged for eight sequential rounds. Finally, bacterial cells were identified by decoding their barcodes and compared with ground truth (Fig. 2c, d and Supplementary Fig. 5). We found that SEER-FISH had excellent precision (median = 0.98, ranging from 0.78 to 0.99) and recall (median=0.89, ranging from 0.61 to 0.97) in taxonomic identification (Supplementary Fig. 5c), as most of the cells were correctly identified. In particular, we found that recall was significantly improved via error correction (27% via 1-bit correction, and 14% via 2-bit correction); otherwise, these observed barcodes would be unidentified because of detection errors (Fig. 2d).

Despite the stringent criteria used in probe design, due to the sequence similarity between closely related bacterial species and complex effects of sequence mismatch on hybridization efficiency, non-specific binding in bacterial rRNA FISH is difficult to be completely eliminated[34]. To investigate how probe non-specificity influenced the performance of SEER-FISH, we systematically profiled the specificity matrix (12 probes vs. 12 bacterial species) using conventional one-step hybridization FISH (Fig. 2e, Supplementary Fig. 6). We found several cases of non-specific hybridization (i.e., off-diagonal fluorescence signal in the specificity matrix), especially for phylogenetically related species. Non-specific binding of probes caused low precision for species PS and AC (due to false positives) and low recall for species PD1 and VA1 (due to false negatives) (Supplementary Fig. 5c, see species list in Supplementary Table 2). Furthermore, we found that the measured fluorescence intensity was in good agreement with the predicted hybridization efficiency[37] (more negative $\Delta G$ means better probe hybridization) (Fig. 2f).

To systematically evaluate the importance of error-robust encoding for sequential FISH with probe non-specificity, we performed simulations for randomly generated codebooks with variable minimal HD (see Codebook generation in Methods). In the simulated experiments, we classified the probe-species pairs into three groups based on binding free energy $\Delta G$, namely specific binding, non-specific binding, and background. In particular, non-specific binding of probes ($-13.0 < \Delta G < -7.3$ kcal/mol) could lead to detection errors. At an intermediate level of non-specificity (Fig. 2g, see Simulations of SEER-FISH in Methods), simulation results showed that error-robust encoding (i.e., increased minimal HD) enabled overall improvements in precision and recall of bacterial identification. Qualitatively similar results were observed in simulations at different levels of non-specificity (Supplementary Fig. 7). In summary, our experimental and computational results showed that sequential FISH with error-robust encoding could tolerate probe non-specificity to provide accurate and sensitive identification of bacterial taxa.

## Profiling the composition of microbial communities by SEER-FISH

To evaluate the reproducibility of taxonomic profiling in microbial communities using SEER-FISH, we performed benchmarking experiments on synthetic communities consisting of 12 species (SynCom12, SynCom12_unequal) and 30 species (SynCom30) (Supplementary Table 2, see Bacterial culture in Methods). Using the R8HD4 codebook (rounds = 8, minimal HD = 4, the barcodes for a specific set of strains (S) = 12, Supplementary Table 5), all 12 species were successfully identified in SynCom12 (Fig. 3a). We found close agreement in the estimated taxonomic composition based on SEER-FISH across different Fields of View and experimental replicates (Pearson correlation R ≥ 0.9) (Fig. 3b). Moreover, we altered the relative abundances of four species in the synthetic community (SynCom12_unequal), where the proportions of FL1 and AD1 were increased to 15.7% and the proportions of AC and PA were decreased to 1%. We found that SEER-FISH accurately quantified changes in the community composition (Fig. 3c).

Furthermore, we profiled a more complex microbial community (SynCom30) to evaluate the performance of SEER-FISH under different encoding schemes. We chose two codebooks, R8HD4 (rounds = 8, minimal HD = 4, S = 30) and R12HD6 (rounds = 12, minimal HD = 6, S = 30) with high F1 scores (i.e., the harmonic mean of precision and recall) predicted by simulations (Supplementary Fig. 8, Supplementary Table 5). All 30 species were successfully identified by SEER-FISH in both R8HD4 and R12HD6 codebooks (Fig. 3d), and the estimated compositional profiles were highly correlated (Fig. 3e, Pearson correlation R = 0.93). In both codebooks, SEER-FISH identified ~80% of the bacterial cells in the community (Fig. 3f). For R12HD6 codebooks, the increase in imaging rounds led to more detection errors, yet its minimal HD allowed for error correction up to 3 bits and a higher F1 score than R8HD4

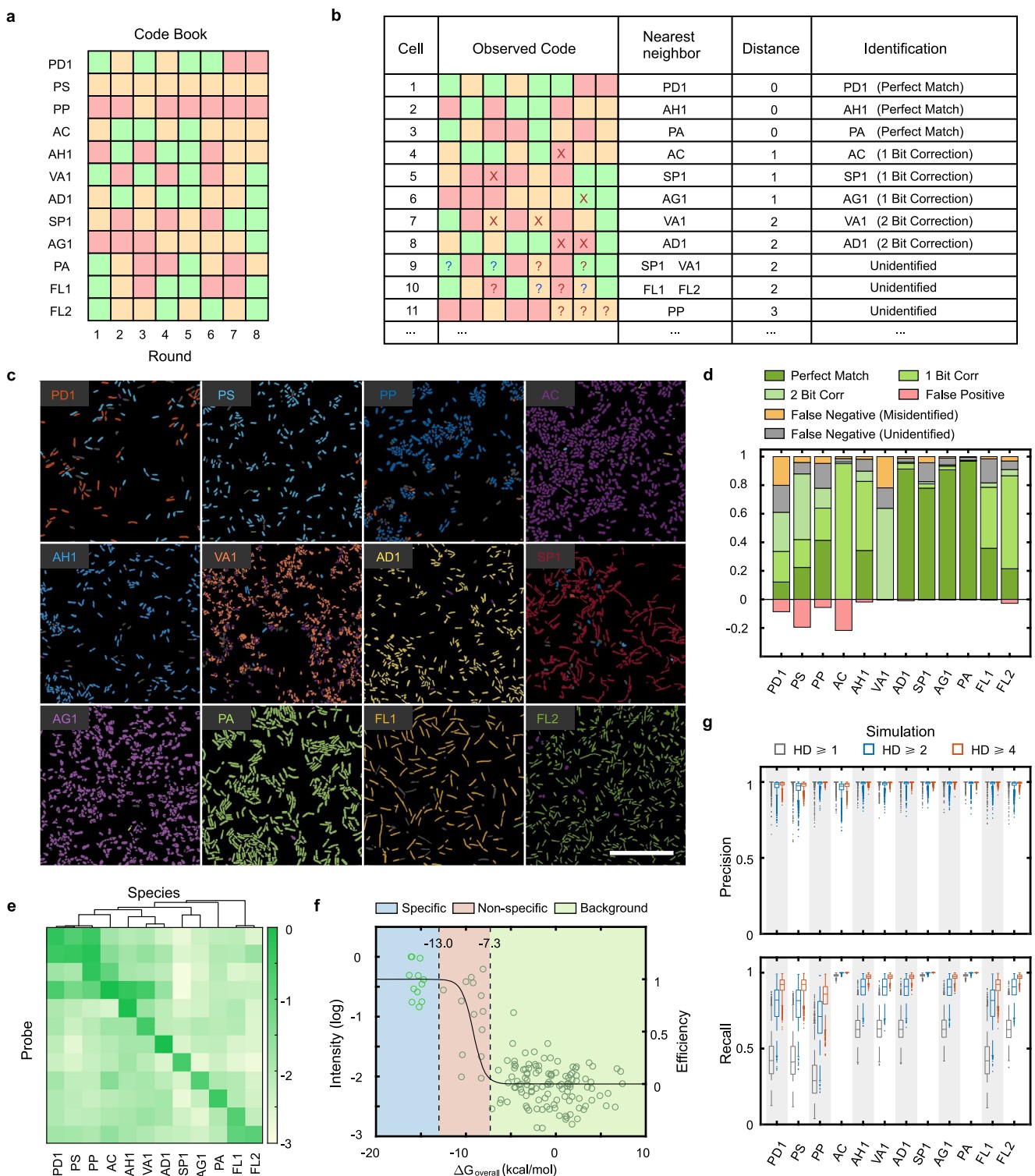

codebooks (Supplementary Fig. 8d). Similar to the simulation results for 12 species (Supplementary Fig. 7), we found that error-robust encoding led to enhanced accuracy in taxonomic identification for the 30-species microbial community (Supplementary Fig. 8). Overall, we found that SEER-FISH can be used to quantify the composition of complex microbial communities and that such profiling is highly reproducible.

## Spatial mapping of microbial biogeography on *Arabidopsis* rhizoplane

Previous studies have shown that the composition and spatial distribution of root microbiome are associated with plant physiology and development[7,38,39]. Rhizosphere microbiomes were found to vary across different root types (e.g., primary and secondary roots) and regions[40,41]. Imaging-based approaches have further confirmed the spatial variation in root-colonized microbes[16,42–44].

To demonstrate the power and usefulness of our method, we implemented SEER-FISH on *Arabidopsis* roots to map the biogeography of microbial communities colonized on the rhizoplane. Axenically grown *Arabidopsis* plants were inoculated with a synthetic community consisting of 12 bacterial species (including ten species isolated from *Arabidopsis* roots[45] and two *Pseudomonas* strains[46]) and co-cultured for 7 days under hydroponic conditions (Fig. 4a).

**Fig. 2 | SEER-FISH enables highly accurate taxonomic identification. a** The codebook used for the validation experiment on the synthetic community consisting of 12 bacterial species (Supplementary Table 5, R8HD4 codebook). **b** Illustration of the decoding scheme for the codebook shown in panel a. Crosses (or question marks) indicate errors that can (or cannot) be corrected by mapping to the nearest neighbor in the codebook. **c** Identification of bacterial species grown in pure culture by eight rounds of imaging (R8HD4 codebook). The pseudocolor of each bacterial species is indicated by its acronym. Scale bar, 25 μm. **d** Quantification of results in panel c. For each species, cells correctly identified (including perfect match, 1-bit correction, and 2-bit correction) are true positives (Green); cells incorrectly identified as other 11 species are marked as misidentified (Orange); cells that cannot be classified to any of the 12 species are marked as unidentified (Gray). Cells of other 11 species incorrectly identified as the corresponding species are false positives (Red). Ratios are normalized by the total cell number of each species. **e** Analysis of probe specificity. The measured fluorescence intensity of bacterial

cells (pure culture, average of ~1000 cells) hybridized with probes designed to target individual species. The species are clustered by the phylogenetic distance between full 16 S sequences (Minimum-Evolution Tree, MEGA-X v10.1.8). Probes (y-axis) follow the same order as the targeted species (x-axis). **f** The relationship between the measured fluorescence intensity and the change in free energy ($\Delta G$) of each probe-species pair. The light and dark green circles indicate the measured fluorescence intensity of diagonal and off-diagonal probe-species hybridization shown in panel e, respectively. The black line indicates the predicted hybridization efficiency $E$ (see Probe design in Methods). **g** Simulations show that both precision and recall of taxonomic identification are improved by error-robust encoding. The colored boxplot indicates the predicted distribution of precision and recall of SEER-FISH with $n = 5000$ randomly generated codebooks ($F = 3$, $R = 8$, $S = 12$) with different minimal HD (HD ≥ 1, 2, 4). The height of the box indicates the first and third quartiles. Source data are provided as a Source Data file.

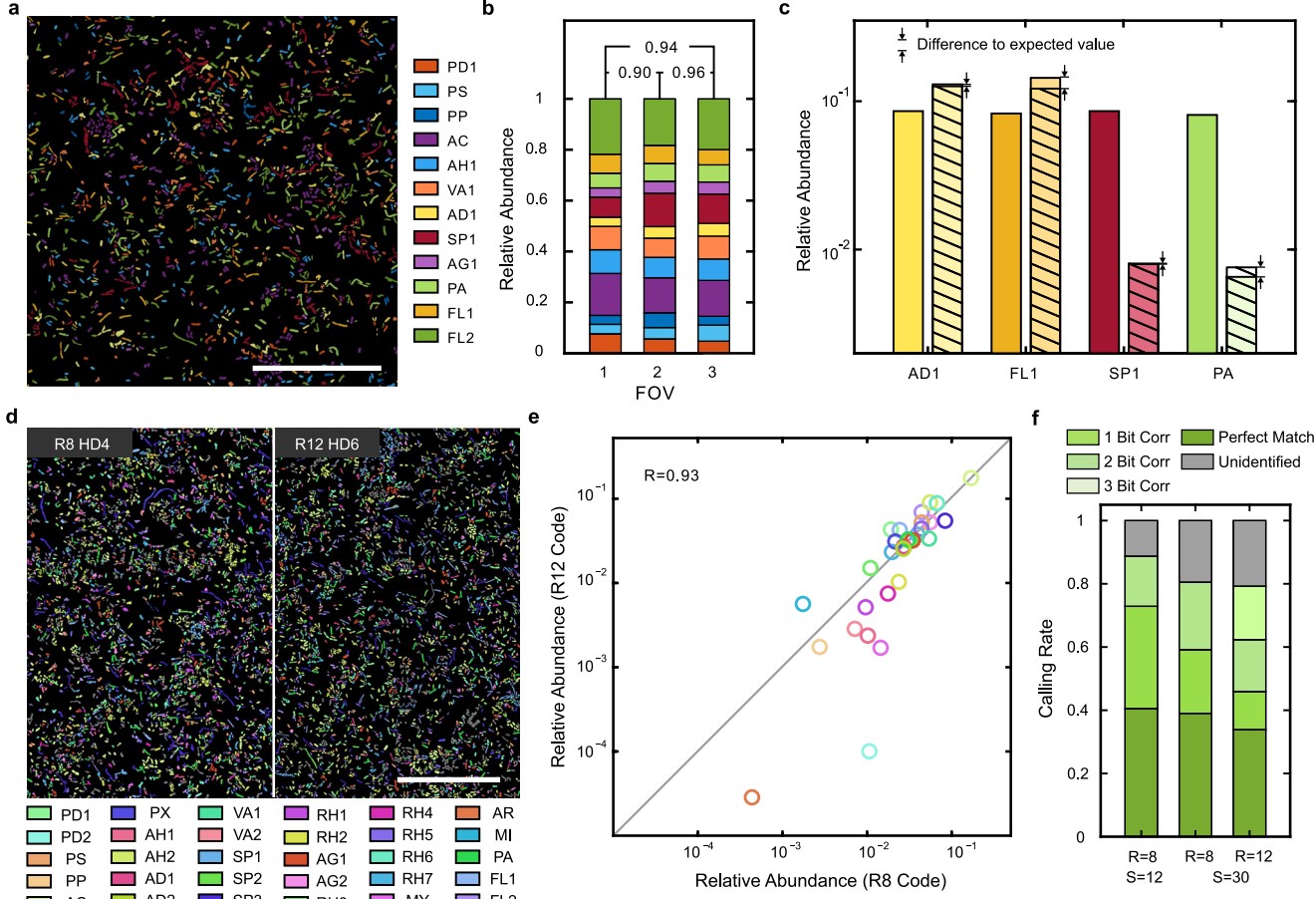

**Fig. 3 | SEER-FISH gives robust estimates of the composition of complex microbial communities. a** Representative image, profiling of SynCom12 based on R8HD4 codebook. Scale bar, 50 μm. **b** Quantification of 12 species relative abundance in SynCom12 in three independent imaging experiments ($n = 15818$, 33365 and 24503 cells, respectively). Pearson correlation between different Fields of View (2 and 3) and between experimental replicates (1 and 2, 1 and 3) are indicated. **c** Bars of the same color indicate the relative abundance of a given species in SynCom12 (left) and SynCom12_unequal (right) quantified by SEER-FISH. The expected relative

abundance after adjustment in SynCom12_unequal is labeled by the stripes. **d** Representative images, profiling of SynCom30 based on two different codebooks (R8HD4, $n = 93{,}596$ cells vs. R12HD6, $n = 101{,}084$ cells). Scale bar, 50 μm. **e** Correlation between the relative abundance profiles estimated by two imaging experiments using codebook 1 ($R = 8$, HD ≥ 4, $S = 30$) and codebook 2 ($R = 12$, HD ≥ 6, $S = 30$). **f** The ratios of identified and unidentified cells. Left bar: SynCom12 (**a**); middle and right bars: SynCom30 (**d**). Source data are provided as a Source Data file.

Root-colonized bacterial cells were detected by the universal FISH probe EUB338, and further validated by nucleic acid staining with SYBR Safe[42,47] (Supplementary Fig. 9). We imaged roots colonized by two different bacterial communities (SynCom13, $n = 3$; SynCom22, $n = 3$). In total, we found that 97.8 ± 1.5% EUB338 labeled cells were labeled by SYBR, and 95.5 ± 2.0% SYBR labeled cells were labeled by EUB338 (Supplementary Fig. 9b). Also, there was no signal for negative

control FISH probe NON338 on bacteria-colonized roots. Thus we demonstrate that bacteria cells colonizing on rhizoplane can be correctly identified by FISH[7,42]. Then we imaged multiple regions on 3 root samples (within ~5 mm to the root tip) and quantified the community composition by SEER-FISH (Fig. 4b–d). Roughly 15% of bacterial cells were lost after eight rounds of imaging (Supplementary Fig. 10). In the regions that we imaged on 3 root samples, a total of ~15,000 bacterial

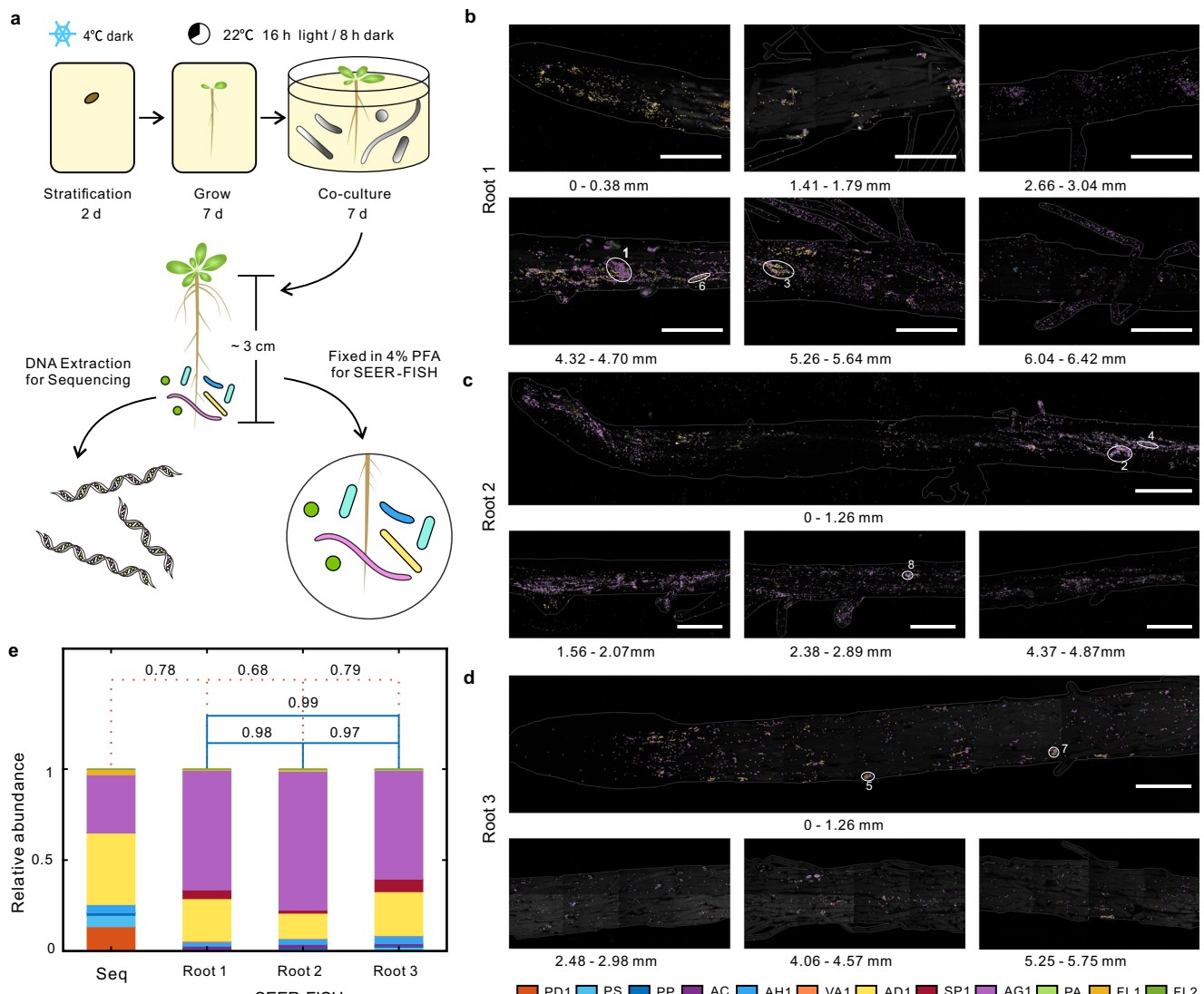

**Fig. 4 | Spatial profiling of microbial communities colonized on *Arabidopsis* roots by SEER-FISH. a** The protocol of synthetic microbial community colonization on *Arabidopsis* roots. *Arabidopsis* seeds were germinated on an MS plate and then colonized by a synthetic community of 12 bacterial species for 7 days under hydroponic conditions. **b–d** The colonization of 12-species synthetic community on different regions of 3 independent roots as identified by SEER-FISH. The edges of the root (white lines) are drawn manually based on the phase-contrast image.

Scale bars, 100 μm. Numbers below each image indicate the distance to the root tip. Circles and numbers indicate the location of bacteria clusters shown in Fig. 5f. **e** The composition of the communities colonized on root measured by 16 S amplicon sequencing (mixture of 10 roots) and SEER-FISH (the Pearson correlation is indicated by orange dash lines). The Pearson correlation between the community compositions on 3 roots measured by SEER-FISH is indicated by blue lines. Source data are provided as a Source Data file.

cells (of 12 species) were successfully identified (including error corrections), and only ~10% cells were unidentified. While we observed variations across different regions (Supplementary Fig. 11), the overall community composition estimated by SEER-FISH were highly similar across 3 roots (Pearson correlation $R > 0.97$, $P < 10^{-5}$). There was also close agreement between the community composition estimated by SEER-FISH and by 16 S rRNA amplicon sequencing of root samples (Fig. 4e).

SEER-FISH allowed us to map spatial patterns of the microbiome along plant roots at single cell resolution. We quantified the distribution of bacterial cells and community composition along the roots (Fig. 5a, b). The imaged regions covered four developmental zones of the root tip[48,49], including the root cap, the meristematic zone, the elongation zone and the differentiation zone (or maturation zone). The two most abundant species, AD1 (*Acidovorax* sp.) and AG1 (*Agrobacterium* sp.) accounted for ~80% of the community (Supplementary Fig. 11). We further analyzed the clustering of bacteria cells using the

linear dipole algorithm[4,50] (Fig. 5c–f, see Clustering analysis in Methods). Indeed, the auto-correlation function revealed that microbes on the root surface clustered at distance up to ~30 μm, which was consistent across 3 root samples and in agreement with visual inspections (Figs. 4 and 5f). Furthermore, we analyzed the pair cross-correlation of the two most abundant species AD1 and AG1 (Supplementary Fig. 12). We found clustering of AD1 and AG1 on root 3, nevertheless, there were substantial variations among the regions that we imaged. The root-to-root variations in spatial patterns suggest stochasticity in bacteria colonization, which should be taken into account in future studies. We identified clusters of bacterial cells ranging from tens to hundreds of square micrometers (Fig. 5e) and found that some clusters consisted of bacterial cells from multiple species (Fig. 5f). Furthermore, we performed contact frequency analysis to identify non-random intertaxon associations[24] (Fig. 5g, see Contact frequency analysis in Methods). Compared to the contact frequency between randomly distributed cells, there were 15 significant spatial associations among 9 species

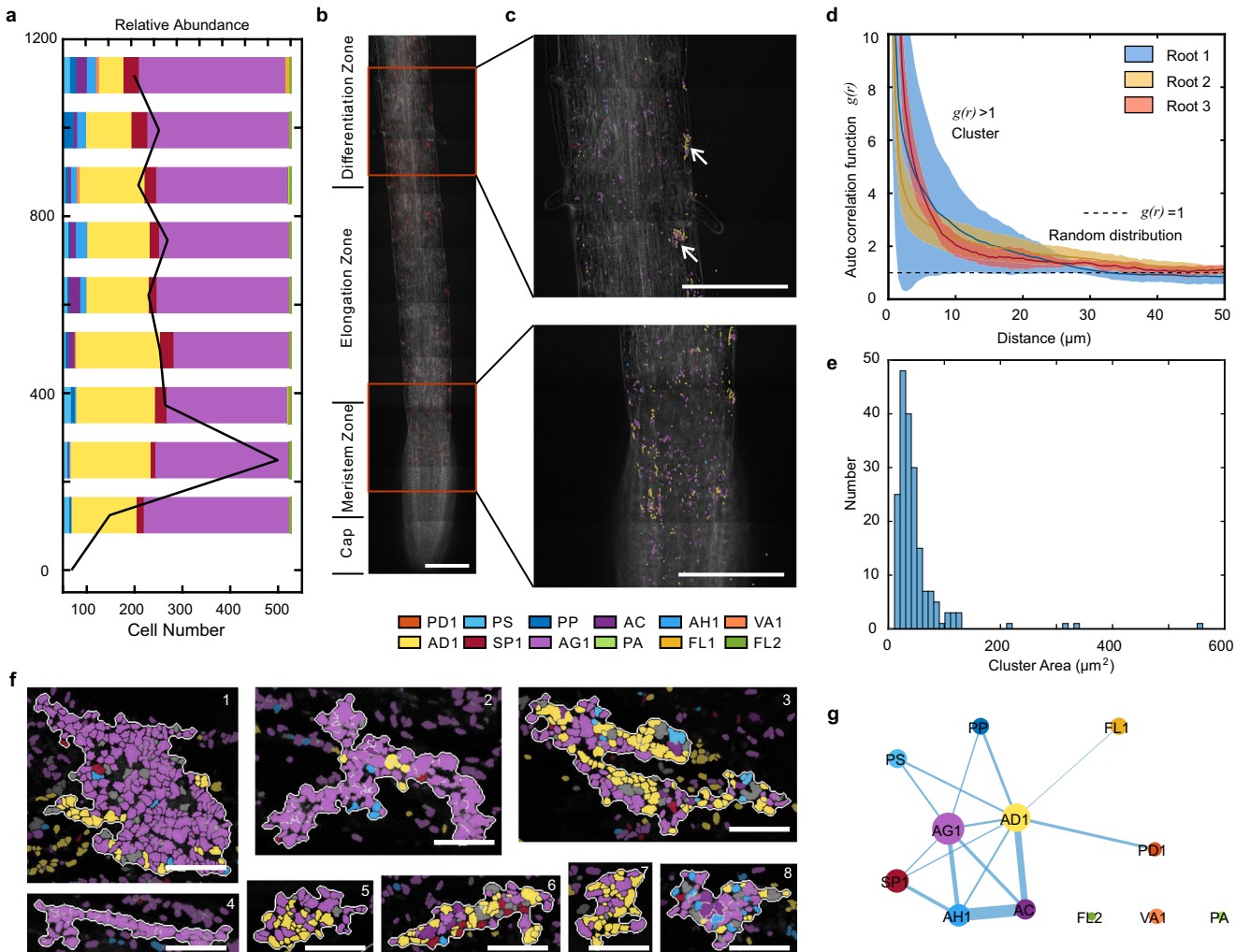

**Fig. 5 | Spatial patterns of the root-colonized microbial communities at single cell resolution. a** Relative abundance of each species (bars) and the number of cells (black line) by imaging along the root of *Arabidopsis*. Each bar indicates the relative abundance in an FOV of 125 μm (width) × 250 μm (length). **b** Representative images illustrate the spatial distribution near the root tip (-1 mm from the tip) labeled by the universal probe EUB338. Scale bar, 100 μm. **c** Bacterial colonization marked by the pseudo color of each species at different regions along the root. Species AG1 and AD1 are found in clusters (white arrows). Scale bar, 100 μm. **d** The spatial correlation of root-colonized bacterial cells is analyzed by linear dipole algorithm (see Contact frequency analysis in Methods). The solid lines indicate the mean auto correlation between bacterial cells, the shadows indicate the 95% confidence intervals estimated by sampling different regions on each root. The horizontal dash line (g(r) = 1) refers to the expected value of a randomized spatial distribution. **e** The distribution of cluster area. **f** Representative images of clusters. Scale bars, 10 μm. The clusters are surrounded by white lines and cells outside clusters are shown with a dimmer color. **g** Intertaxon spatial contacts observed for 12-species bacterial communities colonized on roots. Each edge shows non-random contact between two species (Supplementary Fig. 13, Methods). The width of edges is proportional to the fold increase in contact frequency compared to randomly distributed cells. The size of nodes is proportional to the relative abundance (log-transformed) of each species. Source data are provided as a Source Data file.

colonized on roots (Supplementary Fig. 13). For example, we found non-random cross-correlation and contact frequency between AD1 (*Acidovorax* sp.) and AG1 (*Agrobacterium* sp.). Visual inspections of high-magnification images (Fig. 5f) also showed clustering of AD1 and AG1 cells. Moreover, AH1 (*Achromobacter* sp.) and AC (*Acinetobacter* sp.) cells appeared frequently in clusters and had significant associations with AD1 and AG1 cells.

## Perturbation on the spatial organization of root-colonized microbial communities by plant metabolites

Plant metabolites have been shown to modulate the composition and function of plant-associated microbiome[51–57]. Camalexin is one of the alkaloid phytoalexin produced and secreted by *Arabidopsis* in response to pathogen invasion and has been reported to affect plant-microbe interactions[58,59]. Fraxetin belongs to the group of coumarin and is typically synthesized and secreted by *Arabidopsis* roots under

iron deficiency[60]. Fraxetin is also recognized for its antimicrobial function[52,54].

Here we applied SEER-FISH to study the effects of camalexin and fraxetin on the spatial organization of root-colonized microbial communities (Supplementary Fig. 14a). The 30-strain microbial community (SynCom30.2) spanned the phylogenetic diversity of *Arabidopsis* rhizosphere microbiome and included members that were previously shown to respond to plant metabolites[52,54]. Our in vitro growth experiments also confirmed selective growth modulation of camalexin and fraxetin on members of the community (Supplementary Fig. 14c). For each root, -80 FOVs were captured (within ~4 mm from the tip) (Fig. 6a–c). The compositional profiles given by SEER-FISH imaging (*n* = 10 roots) were in good agreement with the profiles given by 16 S amplicon sequencing (Supplementary Fig. 14b). For camalexin-treated and fraxetin-treated plants, root-colonized microbiota showed clear shifts in composition compared to plants in the control group (Fig. 6c,

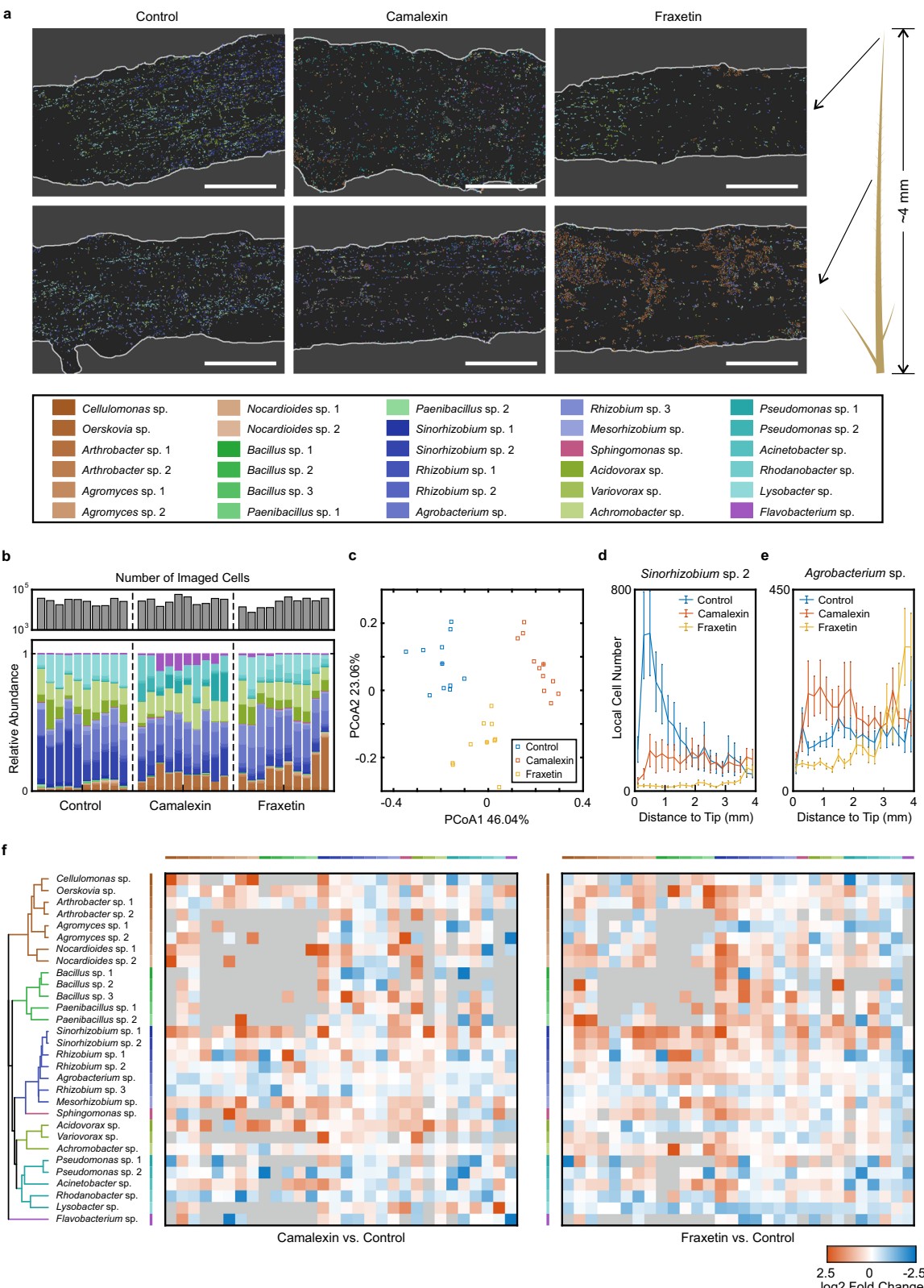

Supplementary Fig. 14c). For example, the abundance of *Mesorhizobium* sp. decreased under camalexin/fraxetein treatment, consistent with the observation that its growth was strongly inhibited by the plant metabolites in vitro.

We examined the spatial distribution of microbial colonization along the root and the perturbations imposed by plant metabolites (Fig. 6d, e and Supplementary Fig. 15). For example, *Sinorhizobium* was

abundant in the control group and mostly colonized the region within 1 mm to the tip. We found that the abundance and the spatial pattern of *Sinorhizobium* strains were significantly altered by camalexin and fraxetin (Fig. 6d, Supplementary Fig. 15c). In contrast, the spatial distribution of *Agrobacterium* sp. was uniform in the control group; in fraxetin-treated plants, *Agrobacterium* sp. showed preferential colonization near the maturation zone (Fig. 6e). These non-uniform and

**Fig. 6 | Perturbation on the spatial organization of root-colonized microbial communities by plant metabolites. a** Representative images of microbial communities in the meristem and elongation zone (-200 µm from the tip, top panels) and in the differentiation zone (-1.7 mm from the tip, bottom panels). Scalebar, 50 µm. **b** The number of imaged microbial cells and the community compositional profiles given by SEER-FISH. The number of cells imaged by SEER-FISH for a root sample in the control group, the camalexin-treated group and the fraxetin-treated group were $2.7 \pm 0.8 \times 10^4$, $3.1 \pm 1.3 \times 10^4$, and $2.5 \pm 1.2 \times 10^4$, respectively. For each experimental group, 10 roots were imaged. For each root, -80 FOVs were captured (within -4 mm from the root tip). **c** Principal Coordinate Analysis (PCoA) based on Bray-Curtis dissimilarity of community compositional profiles given by imaging. Solid square indicates the compositional profile averaged over 10 root samples. **d, e** Spatial distribution of *Sinorhizobium* sp. 2 (**d**) and *Agrobaterium* sp. (**e**) along the root. Error bars are SEMs ($n = 10$ roots). **f** Differential spatial association analysis on root-colonized microbial taxa between camalexin-treated (or fraxetin-treated) plants and control plants (see Spatial association analysis in Methods). Fold change refers to $\log_2$[(association frequency on camalexin-treated (or fraxetin-treated) roots/simulated random frequency in treated roots)/(association frequency on control roots/simulated random frequency in control roots)]. Gray areas indicate that the analysis is not applicable. Source data are provided as a Source Data file.

taxon-specific spatial patterns of root-colonized microbes indicate strong heterogeneity in root environments (e.g. region-specific exudates) as well as diverse microbial traits[43,44,61]. Furthermore, we found that plant metabolites disrupted the spatial associations between several bacterial taxa (Fig. 6f and Supplementary Fig. 16, see Spatial association analysis in Methods). Camalexin treatment significantly increased spatial association between *Acidovorax* sp. and *Arthrobacter* sp. 2 (Supplementary Fig. 16a), while fraxetin treatment altered spatial associations between *Lysobacter* sp. and *Sinorhizobium* strains (Supplementary Fig. 16b). Taken together, we demonstrate the utility of our methods in host-associated microbiome samples and highlight the need for comprehensive analysis of spatial heterogeneity in microbial communities (e.g. colonization on plant roots).

## Discussion

In this study, we demonstrated that SEER-FISH is a highly multiplexed and accurate imaging technique for investigating the spatial organization of microbiomes. We developed an iterative hybridization and imaging method for microbiome samples. Using error-robust encoding schemes, we showed that SEER-FISH provided accurate spatial profiling of microbial communities. The application of multiplexed FISH methods will greatly facilitate the understanding of biogeography of host-associated bacteria communities at single-cell resolution. The superior multiplexity of sequential labeling in SEER-FISH combined with the two-step labeling probe design of HiPR-FISH can be used in future studies to take advantage of these complementary approaches, reducing probe costs and pushing spatial mapping of microbiomes to new frontiers. One exciting prospect is to profile meta-transcriptomes in situ, as simultaneous labeling of mRNA and rRNA is feasible for single bacterial strains[14]. The labeling strategy of HiPR-FISH requires a high abundance and uniform distribution of microbial RNA, but this is not necessarily required for SEER-FISH. In addition, the complexity of meta-transcriptomes ($> 10^7$ genes in human gut microbiomes) require an increase in multiplexity, which can be achieved by sequential labeling. The multiplexity of SEER-FISH can be readily extended by increasing the number of fluorophores and the rounds of imaging. While only three fluorophores were used in this study, our method can easily incorporate more colors and spectral imaging[24,28].

Recently, a sequential FISH method reported as par-seqFISH spatially profiled the expression of -100 marker genes in bacterial populations of *Pseudomonas aeruginosa* at single-cell resolution[62]. par-seqFISH focused on spatial transcriptomics within a bacterial population (of one species), while SEER-FISH was developed to study spatial metagenomics of a multi-species microbial community. In par-seqFISH, mRNAs were labeled once with a nonbarcoded approach (i.e., the multiplexity scales linearly with the number of imaging rounds); while in SEER-FISH, rRNAs were labeled repeatedly with error-robust encoding (i.e., the multiplexity scales exponentially with the number of imaging rounds). Inspired by the dimensionality reduction approaches commonly used in single cell transcriptomics analysis, we visualized the multi-round, multi-color SEER-FISH imaging data in dimension reduced maps by t-SNE (Supplementary Fig. 17). For both simulated and real imaging data, we found that bacterial cells of the same species were clustered in the dimension reduced map,

and different species were clearly separated. In future studies, the dimensionality reduction approach may be used to identify unknown microbial taxa.

The incorporation of error-correction strategies, originally implemented in MERFISH for multi-round mRNA profiling[32,33], is expected to improve the accuracy of target identification, but has not been studied in the context of microbiome. By incorporating error-robust encoding schemes in SEER-FISH, we show that the precision and recall of taxonomic identification can be improved, particularly in scenarios where non-specific hybridization is unavoidable. In mRNA labeling, non-specific calling is less common than dropout errors[32,63]. Thus, in MERFISH, modified HD4 codes (with only four "1" bits in 16-bit barcode) were used to minimize dropout errors ($1 \to 0$). In contrast, detection errors in bacterial rRNA FISH are mainly caused by non-specific (i.e., off-target) labeling of phylogenetically related rRNA sequences; dropout errors $1 \to 0$ are negligible due to the high abundance of rRNA. While non-specific binding cannot be completely avoided for rRNA FISH probes (the target region for probe design is limited), the fluorescence intensity of the specific probe is on average much higher than nonspecific probes (Fig. 2f). Therefore, in our image analysis, the color code of each cell in each round was determined by the brightest fluorescence channel in the corresponding round (see Image analysis in Methods). Because $F \to F'$ errors can be better avoided than $0 \to F'$ errors, we chose to exclude the non-fluorescent code in the codebook (i.e., color code = 0) to minimize detection errors caused by non-specific labeling (color code $0 \to F'$). For a given set of FISH probes, codebooks can be optimized to account for non-specificity of probes and improve precision and recall. Here we provide an illustration of codebook optimization and its practical use in design of barcodes. We used the measured probe specificity (Supplementary Fig. 6) to predict the F1 score of codebooks for the 12-species synthetic community SynCom12. The codebook that we used for imaging SynCom12 had high predicted F1 score (predicted F1 score = 0.92); we found that the fraction of unidentified cells was $0.11 \pm 0.01$ in experiments, consistent with simulation results (Supplementary Fig. 18). In comparison, we randomly picked a codebook with low predicted F1 score (predicted F1 Score=0.12) and performed an independent imaging experiment on SynCom12; we found that the fraction of unidentified cells increased to $0.19 \pm 0.01$, and the recall rate of PD1 species was substantially lower (Supplementary Fig. 18a–c). To label a community with 12 targets, there are more than $6.7 \times 10^{20}$ sets of R8HD4 (S = 12) possible codebooks, which cannot be enumerated for evaluation. Thus, we used a genetic algorithm to optimize codebooks of SEER-FISH to achieve high F1 score (Supplementary Fig. 18d, e, see Codebook optimization in Methods). Given the information of non-specificity of FISH probes (experimentally measured or predicted), this computational approach can guide the design of error-robust encoding schemes. Other experimental modifications to reduce non-specific hybridization, such as increasing hybridization stringency[15], adding competitor probes for off-target taxa[28] or dual probes with overlapping specificity[34], can also be used to improve accuracy and are readily compatible with SEER-FISH.

Depending on the samples, particularly for complex communities, the workflow of SEER-FISH can be improved in several aspects.

Firstly, to improve the hybridization efficiency of FISH probes, samples can be pretreated with lysozyme to increase the permeability of cells (especially for gram-positive bacteria)[7] and/or incubation with high-concentration formamide for thorough denaturation of rRNA. With the optimizations above, we applied SEER-FISH to image a highly complex community composed of 130 strains colonized on *Arabidopsis* roots (Supplementary Fig. 19). The 130 strains were grouped into 90 target taxa based on the similarity of 16 S rRNA sequences. Similar to the validation experiments that we performed on 12 taxa (Fig. 2 and Supplementary Fig. 5), we used pure cultures of 90 taxa to evaluate the performance of taxonomic identification by SEER-FISH in highly complex communities (Supplementary Fig. 19a–c). We found excellent precision (median = 0.87) and recall (median = 0.78) for most taxa. As a proof-of-concept experiment, we used these 90 FISH probes to image a synthetic community of 130 strains colonized on *Arabidopsis* root (Supplementary Fig. 19d, e) and correctly identified ~65% bacterial cells. Further improvement in probe design would be helpful in the application of multiplexed FISH methods to profile highly complex communities. Secondly, the segmentation algorithm that we used is potentially limited to samples with a clean background. Imaging more complicated samples or dense microbial communities (e.g., biofilms) may require improvements in the image analysis workflow. Thirdly, a general nucleic acid dye could be incorporated in multiround imaging to label bacterial cells with a uniform signal[42].

The application of SEER-FISH on plant samples has revealed the micron-scale spatial organization of root-colonized microbial communities, including clustering of multiple species and intertaxon spatial associations. The clustering of bacterial cells colonized on plant surface has been previously reported[4,42,64]. Clusters can form via the growth of microcolonies upon successful colonization[65]. Formation of clusters on plant surface may be critical for bacterial fitness under environmental stress[66]. For example, it has been proposed that phyllosphere bacteria form clusters to deal with desiccation stress[64]. Preferential attachment is another potential mechanism for the formation of clusters, as previous studies have shown co-localization of immigrant and resident bacterial cells[67]. Our observation of multiple bacterial species in clusters may lend support to the hypothesis of preferential attachment[67,68]. Furthermore, the micron-scale intertaxon spatial associations may be indicative of short-range interactions[69] (e.g., quorum sensing, metabolic cross feeding, niche competition, contact-dependent inhibition) and will guide mechanistic studies on the ecology of complex microbial communities. Lastly, alterations in the spatial structure of microbiome during host development[16], stress response (biotic and abiotic)[70,71] and diseases could lead to novel insights in host-microbiome interactions. Root-secreted metabolites have been found to regulate the composition of rhizosphere microbiome, but their effects on the spatial organization of microbiome remain largely unknown. Future investigations along these lines will deepen our understanding of microbiome assembly in rhizosphere/phyllosphere and its implications to plant fitness[72].

Finally, we envision that the integration of SEER-FISH with other spatially resolved technologies will have broad impacts on microbiology/microbiome research. For example, SEER-FISH can be combined with expansion microscopy to profile the transcriptome of single bacterial cells[73]. Together with mass spectrometry imaging[74] or multiplexed protein maps[75], SEER-FISH can unravel the functions of complex microbial communities in space and their interactions with the host at the molecular level.

## Methods

### Bacterial culture

A full list of bacterial strains used in this study is included in Supplementary Table 2 and Supplementary Table 3. Strains isolated from *Arabidopsis* root microbiota were kindly provided by Professor Paul Schulze-Lefert at the Max Planck Institute for Plant Breeding Research.

*Pseudomonas* strains PP (WCS358) and PS (WCS417) were kindly provided by Professor Corné Pieterse at Utrecht University. All strains were cultured in ½ Tryptic Soy Broth (TSB) medium (HuanKai Microbial, 024051) (28 °C, shaken at 200 rpm under a normal aerobic atmosphere) and harvested at ~0.8 $OD_{600}$ (BioTek, Synergy H1 Hybrid Multi-Mode Reader). Bacterial cells were centrifuged at $5000 \times g$ for 5 min, resuspended and washed with 1× PBS (Boster, AR0030). For fixation, cells were resuspended and incubated in 4% paraformaldehyde (DF0135-2; Leagene) at 4 °C for 3 h, followed by washing with PBS and resuspension in 50% ethanol (for permeabilization of gram-positive cells)[76]. Then cells in 50% ethanol were stored at −20 °C before FISH imaging. For the synthetic community of 12 bacterial strains, two different compositions were created: 1) SynCom12 (mixed with equal $OD_{600}$): 8.3% for each strain. 2) Syncom12_unequal (mixed with unequal $OD_{600}$): the proportions of *Flavobacterium* sp. (FL1) and *Acidovorax* sp. (AD1) were 15.7%; the proportions of *Acinetobacter* sp. (AC) and *Paenibacillus* sp. (PA) were 1%; and the proportions of other strains were 8.3%. For the synthetic community of 30 bacteria strains (SynCom30), all strains were mixed at equal $OD_{600}$.

### Probe design

All probes used in this study are listed in Supplementary Table 3. Oligonucleotide probes were conjugated with three different types of fluorophores at the 5' terminus: FAM, Cy3, and Cy5 (ordered from General Biol). Oligonucleotide probes were designed to target 16 S rRNA or 23 S rRNA using a custom pipeline (Supplementary Fig. 4). rRNA sequences of the bacterial strains used in this study (Source Data) were extracted from whole genome sequencing data using Prokka[77], built into a local database and imported to the ARB program (www.arb-home.de). The 'Probe Design' function of ARB was used (parameter settings: 18–21 nucleotides, 45–60% GC content)[36]. Probes with fewer than three mismatches to non-target sequences were excluded. The change in free energy ($\Delta G$) for probe-target binding was calculated by mathFISH[37] and probes with $\Delta G < -13.0$ kcal/mol were chosen as candidate probes. Alternatively, probes can also be designed by DECIPHER[34] (Supplementary Table 3). The database of aligned 16 S rDNA sequences with phylogenetic group information were imported and using DesignProbes function in DECIPHER package for probe design.

The hybridization efficiency $E$ is predicted as (Fig. 2f):

$$E = \frac{e^{(-\Delta G/RT)}C_p}{1 + e^{(-\Delta G/RT)}C_p} \tag{1}$$

where $C_p$ is the probe concentration of, $R$ is the molar gas constant and $T$ is the temperature.

### Codebook generation

The codebook is named according to the number of imaging rounds (R), colors (F), and the minimal Hamming distance (HD). All codebooks used in this study contain three colors ($F = 3$); the color code equals 1, 2, and 3 for FAM, Cy3 and Cy5 fluorophores, respectively. First, all $F^R$ barcodes for R rounds of hybridization were generated. Then, the error-robust codebook was generated by repeated removal of barcodes whose distance to the seed barcode was less than the specified minimal HD. Each barcode in the codebook was taken as the seed code until the distance between any two barcodes in the codebook was equal to or larger than the specified minimal HD. The barcodes for a specific set of strains (S) were randomly drawn from the codebook. The codebooks used in the experiments of this study are included in Supplementary Table 5.

### Coverslip functionalization

Coverslips (40-mm, #1.5; Bioptechs) were immersed in potassium dichromate concentrated sulfuric acid cleaning solution for 2 h,

washed with water and then rinsed with distilled water more than three times, soaked in 95% ethanol for 12 h, and then air-dried. 50X Anti-Slice Escaping Agentia APES (Sangon Biotech, E676003) was diluted 1:50 with acetone and the prepared working solution was used immediately. The coverslips were dipped into the freshly prepared working solution for 20–30 s, then removed, followed by a pause before they were washed three times with distilled water to rinse unbound APES. Adhesive processed coverslips were put in a dust-free environment and kept dry.

## Multi-round FISH imaging

An adhesive coverslip coated with sample was assembled into a Bioptechs FCS2 flow chamber with temperature control (Supplementary Fig. 1b). Fixed samples were first adhered onto adhesive coverslips (40 mm round, 0.15 mm thick); then a silicone gasket (40 mm round, 0.75 mm thick) with a central rectangle cavity was placed on the coverslip; and then the micro-aqueduct slide was placed on the gasket. The coverslip, gasket and micro-aqueduct slide constitute a sandwich structure. Samples were in the cavity where buffers passed through. Fluidics was controlled via a peristaltic pump (LongerPump, BT100-2J) and set at a constant flow velocity of 500 μl/min (10 rpm). Buffers were warmed by a metal bath at 50 °C to get rid of dissolved oxygen before passing through the flow chamber. Multiple rounds of probe hybridization, imaging, and probe dissociation were performed as follows (Supplementary Fig. 1a):

1) Probe hybridization and washing: 1 mL of hybridization buffer (0.9 M NaCl, 0.02 M Tris-HCl (pH 7.6), 0.01% SDS, and 20% formamide (Aladdin, F103362)) with probes was flowed through the sample for 2 min and then incubated for 3 min at 46 °C. Probes used in each round were determined by the codebook. Samples were washed by washing buffer (0.215 M NaCl, 0.02 M Tris-HCl (pH 7.6), 0.01% SDS, 5 mM EDTA) for 2 min at 46°C to eliminate residual and nonspecific binding of the probes.

2) Imaging: Images were acquired by a confocal laser scanning microscope (Nikon, A1) with a Plan Apo λ 100 x oil objective lens (Nikon, 1.45 NA). Multiple (~4–36) fields of view (125 μm by 125 μm) were collected by sequential excitation with laser lines 488 nm, 561 nm, and 640 nm. Phase-contrast images were acquired by a transmitted detector using a 640 nm laser. The constant focus during imaging was achieved by Nikon PFS autofocus. The image acquisition settings are listed in Supplementary Table 4.

3) Probe dissociation: dissociation buffer (70% formamide, 0.02 M Tris-HCl (pH 7.6), 0.01% SDS, 5 mM EDTA) was flowed through the samples at 46 °C for 2 min to strip off hybridized probes.

## Image analysis

First, phase contrast images of each round were used for alignment. The images were aligned to the position with maximum cross correlation by custom MATLAB scripts to eliminate the position shift during multiple rounds of imaging. Then, the phase contrast image (for in vitro bacterial communities) or inverted fluorescence image with the universal probe EUB338 (for root-associated bacterial communities) were segmented into binary mask images using an adaptation threshold[78] followed by the watershed algorithm[79] (Supplementary Fig. 2). Finally, the fluorescence intensity of each cell was obtained with the mask generated by image segmentation. The color code of each cell in each round was determined by the brightest fluorescence channel in the corresponding round (color codes equal 1, 2, or 3 for FAM, Cy3, or Cy5 channels, respectively). If the fluorescence intensity was not significantly brighter than the background in all fluorescence channels ($p > 0.05$, t-test), the code of the corresponding bacterial cell was marked as 0. A bacterial cell is labeled as "lost" if it is not detected in any fluorescence channel for more than 3 rounds (Supplementary Fig. 10).

## Barcode identification

The barcode of each bacterial cell was mapped to the codebook (Supplementary Table 5) to find the nearest neighbor. For barcodes with minimal HD of 2k ($k = 1, 2,...$), if the observed barcode was the same as (perfect match) or had less than n-bit difference (n-bit correction, n < k) from the nearest neighbor, the bacterial cell was successfully identified. If the observed barcode had k-bit difference from the nearest neighbor in the codebook and the nearest neighbor was unique, the cell was also successfully identified (k-bit correction). Otherwise, the cell was labeled as unidentified. For example (Fig. 2b), if there was a 1 bit difference between the observed barcode and the nearest candidate barcode, the corresponding cell was identified by 1-bit correction. If there were 2 bit differences between the observed barcode and the nearest candidate barcode, the error could be corrected only when there were 3 or more bits of difference for other candidate barcodes. The cell was marked as unclassified (unidentified) for other conditions.

## Precision and recall calculation

The precision and recall of each bacterial species are calculated by the following equations:

$$Precision = \frac{TruePositive}{TruePositive + FalsePositve} \tag{2}$$

$$Recall = \frac{TruePositive}{TruePositive + FalseNegative} \tag{3}$$

F1 score is calculated as the harmonic mean of precision and recall

$$F1 = \frac{2 \cdot Precision \cdot Recall}{Precision + Recall} \tag{4}$$

## Imaging of bacterial communities on *Arabidopsis* roots

Surface sterilized *Arabidopsis* seeds of wild-type (Col-0) were sown on 1× Murashige and Skoog (MS) medium (Solarbio, M8520) with 3% sucrose and 0.6% agar. After 2 days of cold-stratification at 4 °C under darkness, the plates were then kept in a growth chamber (22 °C, 16 h light/8 h dark, 50% humidity) for 7 days. Meanwhile, bacteria to be inoculated were pre-cultured with ½ TSB respectively as previously described. Cells were collected by centrifuging and then washed with 1× PBS. The OD$_{600}$ of each strain was determined with the BioTek plate reader. The synthetic bacterial community (OD$_{600}$ = 0.01) containing all strains at equal proportions was inoculated in 1× MS liquid media (for treatment groups, camalexin or fraxetin was introduced at concentration of 100 μM). 7-day-old (for SynCom12 and SynCom130) or 5-day-old (for SynCom30.2) seedlings were transferred into 12-well or 6-well culture plates using sterilized tweezers to co-culture with bacterial communities. After 7 days (for SynCom12 and SynCom130) or 5 days (for SynCom30.2) of bacteria-plant co-culture, the roots of seedlings were fixed with 4% paraformaldehyde (PFA) in 1× PBS at 4 °C for 3 h, then rinsed and stored in 50% ethanol at −20 °C. Then samples were taken out and assembled into FCS2 flow chamber before SEER-FISH imaging. In particular, for imaging Arabidopsis root colonized with SynCom130 or SynCom30.2, fixed roots in FCS flow chamber were pretreated for the improvement of probe hybridization efficiency before SEER-FISH imaging. To increase the permeability of cells, sample was flowed by lysozyme solution (10 mg/mL) for 2 min and with 3 min incubation at 37 °C. After PBS washing, samples were flowed by 85% formamide for 2 min and with 8 min incubation at 46 °C for thorough denaturation of rRNA. These sample pretreatments greatly improved the probe hybridization efficiency and the concentration of each target probe is decreased to 15 nM accordingly, which avoids background residual caused by the high total concentration of all probes in each round. Multi-round FISH imaging was carried out as

described above. All bacteria were labeled with the universal bacterial probe EUB338 during the first round of imaging. The concentration of each target probe in each round was 250 nM for SynCom12 root samples, and reduced to 15 nM for SynCom30.2 and SynCom130 root samples. The duration of probe hybridization was 8 min for SynCom12 root samples, and extended to 15 min for SynCom 30.2 and SynCom130 samples. We extended the hybridization time to account for potential effects of reduced probe concentration on hybridization efficiency.

## 16 S amplicon sequencing of synthetic bacterial communities

Roots of *Arabidopsis* were harvested, rinsed in 1× PBS buffer and stored at −80 °C until further analysis. Each sample of ten frozen roots was treated in liquid nitrogen briefly before manual homogenization using plastic pestles. Genomic DNA was extracted with the DNeasy Plant Mini Kit (QIAGEN GmbH, 69106). The sample was amplified with barcode primers 799 F (5′AACMGGATTAGATACCCKG-3′) and 1193 R (5′ACGT-CATCCCCACCTTCC-3′) against the V5-V7 region of the bacterial 16 S rRNA gene. The 25 μl PCR reaction contained 12.5 μL of 2× PrimeSTAR Max Premix (TaKaRa, R045A), 0.4 μM of each primer (Genewiz), and 1 ng genomic DNA. PCR conditions were set as: 1) 98 °C for 5 min; 2) 35 cycles of 98 °C for 10 s, 50 °C for 15 s, and 72 °C for 15 s; 3) elongation at 72 °C for 5 min. PCR products were purified using a Gel Extraction Kit (OMEGA, D2500-01) and quantified with Qubit 4 fluorometer (Invitrogen). The amplicon library was sequenced by Illumina MiSeq (500-cycle, V2 kit). Raw 16 S rRNA amplicon sequence data were processed by QIIME2[80]. The forward and reverse reads were merged by VSEARCH[81]. Reads were demultiplexed and aligned to a reference set of 16 S amplicon sequences to calculate the relative abundance of each taxon using custom Python scripts.

## Simulations of SEER-FISH

The intensity of each bacterial cell of species $i$ in channel $k$ at each round is

$$I_{ik} = \sum_{p=1}^{n} P_{pi} C_{pk} \tag{5}$$

$P_{pi}$ is the fluorescence intensity when probe $p$ is hybridized to species $i$. $C_{pk}$ (=1 present, 0 absent) indicates the code status of probe $p$ in fluorescence channel $k$. The strength of hybridization was assumed to be the same in different fluorescence channels and classified into three groups (specific binding, non-specific binding, and background), according to the overall Gibbs free energy change (ΔG) when probe $p$ hybridized to species $i$. The log-transformed fluorescence intensity ($\log(P_{pi})$) is drawn from a normal distribution. The mean log-transformed fluorescence intensity is set to −0.3, −0.9 and −2 for specific binding, non-specific binding and background, respectively (Fig. 2g, Supplementary Fig. 6c. The standard deviation of $\log(P_{pi})$ was set as 0.3 (Supplementary Fig. 6d). For simulations shown in Supplementary Figs. 7, 8, the mean log-transformed fluorescence intensity for non-specific binding was set to −0.6 for strong non-specificity, −1.2 for weak non-specificity. The fluorescence intensity of each cell in each fluorescence channel was drawn independently during each round according to the codebook, and the barcode was identified as described above.

To predict the F1 score with measured probe specificity, the log-transformed fluorescence intensity ($\log(P_{pi})$) is drawn from a normal distribution

$$f(x) = \frac{1}{\sigma\sqrt{2\pi}} e^{\frac{1}{2}\left(\frac{x-\mu}{\sigma}\right)^2} \tag{6}$$

where $\mu$ and $\sigma$ is set as the mean and standard deviation of the log-transformed intensity when probe (with corresponding fluorophores) $p$ is hybridized to species $i$.

## Codebook optimization

We used the genetic algorithm to optimize codebooks to achieve high F1 score. Briefly, for the first round, 200 sets of codebooks were randomly generated and 15 codebooks with highest F1 score were selected. For each codebook selected from the previous round, 12 mutated codebooks were generated by random shuffling of the barcode for one of the 12 species. The 15 selected codebooks from the previous round, 180 mutated codebooks, and 5 randomly generated codebooks, were evaluated together to select for 15 codebooks with highest F1 score for the next round. This process can be repeated until the codebook with desired F1 score has been found.

## Clustering analysis

The clustering of bacterial cells is analyzed by the linear dipole algorithm as previously described[4,50]. Briefly, the pair cross-correlation function between two bacterial populations is defined as

$$g(r) = \frac{P_r}{2D_i D_j} \tag{7}$$

where $P_r$ is the probability for a dipole with length $r$ hit the bacteria from the two populations. $D_i$ indicated the density of population $i$. To analyze bacterial cells belonging to the same population, the correlation function is defined as

$$g(r) = \frac{P_r}{D^2} \tag{8}$$

where $D$ is the density of the population of interest. If bacterial cells are randomly distributed at distance r, g(r) =1; in contrast, if bacterial cells clustered at distance r, g(r) > 1.

For each image, a mask of the root is manually drawn as the region of interest based on the phase-contrast image. Linear dipoles are evaluated only if both ends fall into the masked regions, therefore, bacterial cells not on the root surface are excluded from the analysis. The cell density is calculated as the area cover by the bacteria cells divided by the area of the root.

## Contact frequency analysis

For each pair of bacteria species, contact events between cells from each species were counted. The contact frequency was calculated as the number of contact events divided by the area of the root. In simulation s, cells were randomly placed and the density of each species was determined by the measured density on root samples. Model cells were simulated as an ellipse, and the single-cell images of each bacteria species were analyzed to determine the length of major and minor axes of ellipses.

## Spatial association analysis

The spatial association between two taxa was calculated as the number of co-occurrence events inside the association range of 10 μm *in planta* compared to a randomly generated co-occurrence test[28]. The test was done by randomly assigning the taxa in the same measured images while keeping the abundance unchanged. For each root, 100 random tests were performed, and the association was calculated as the fold change between the measured co-occurrence and the mean value of random simulated tests. If the measured co-occurrence events were too low, the corresponding measurements are dropped. The spatial association analysis was performed for each experimental group, and differential spatial association was calculated as the fold change between the treatment group and the control group. The statistical significance was determined by independent t-test and the significance threshold level was adjusted by Bonferroni correction. The number of hypotheses tested was calculated as the number of unique taxon−taxon associations detected across all roots.

## Statistics & reproducibility

No statistical method was used to predetermine sample size. No data were excluded from the analyses. Statistical significance was determined by unpaired one-side or two-side Student t-test. Correlation coefficient was calculated using Pearson correlation or Spearman correlation. $P$-value < 0.05 was considered as statistically significant. Bonferroni correction was used for multiple hypotheses testing. For repeated experiments and biological replicates, the exact number is indicated in the figure legends. For results shown in Fig. 2c and Supplementary Fig. 1c, 3, 19, representative images and analysis was based on a single imaging experiment.

## Reporting summary

Further information on research design is available in the Nature Portfolio Reporting Summary linked to this article.

## Data availability

The raw data generated in this study have been deposited in Zenodo [https://doi.org/10.5281/zenodo.5100490]. Source data are provided with this paper.

## Code availability

The source code used for SEER-FISH image analysis have been deposited in Github [https://github.com/JacobZuo/SEER-FISH/] and achieved at Zenodo [https://doi.org/10.5281/zenodo.7656027][82].

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

## Acknowledgements

We thank Prof. Yang Bai (Institute of Genetic and Developmental Biology, Chinese Academy of Sciences) for help with strain shipping and cultivation, plant experiments and critical comments on the manuscript. We thank Prof. Paul Schulze-Lefert for sharing the strains. We thank Dr. Ziyu Li for technical advice and assistance with instrumentation. We thank Prof. Ancheng Huang, Prof. Jonathan Friedman, Prof. Chao Zhong and members of the LD lab for insightful discussions. We thank Ke Yu, Yuqian Wu, Yang Bai (SIAT), Sihong Li for their help during the early stage of this project. This work was supported by grants National Key R&D Program of China 2019YFA0906700 (L.D.), Natural Science Foundation of China 31971513, 32061143023 (L.D.), Natural Science Foundation of Guangdong Province of China 2022A1515011513 (W.Z.), National Natural Science Foundation of China 32100072 (W.Z.) and Shenzhen Engineering Research Center of Therapeutic Synthetic Microbes XMHT20220104015 (F.J.).

## Author contributions

L.D. conceived and supervised the study. L.D., Z.C., and W.Z. designed the experiments. Z. C. set up the experimental protocol and performed the imaging experiments. W.Z. performed data analysis and simulations. L.W. performed the plant experiments. Z.Q. assisted with imaging experiments. J.C. assisted with bioinformatics analysis. F.J. assisted with instrumentation setup. L.D., Z.C., and W.Z. wrote the manuscript. All authors discussed the results and commented on the manuscript.

## Competing interests

The authors declare no competing interests.
