## [Peer Review File · Nature Communications]

Reviewers' Comments:

Reviewer #1:

Remarks to the Author:

I want to congratulate the authors on this wonderful method and its development. I am fully supportive of the publication of this manuscript albeit I still have minor comments and suggestions for the authors.

The authors have devised a way that allows them to rapidly hybridise bacteria up to three FISH probes at a time and repeat this process "n" times. Together with an automatic microscopy setup, this allows the error-robust identification of bacterial taxa.

The technique demonstrated in the manuscript holds tremendous potential. The study is barely scratching the surface of the potential utilisation of the method. I feel that the title is slightly overselling the spatial aspect of the study. Although the authors have used their method to profile the distribution of synthetic communities across the length of the tips of an Arabidopsis root, which is truly awesome. Their technique and actually provided images provide an already more detailed insight into the distribution of individual cells on Arabidopsis roots - the spatial relationship between different biomasses at the micrometer resolution can be investigated. Similar approaches albeit less in depths have not been attempted by many, the authors are citing only one of these studies and make no attempt to analyse their data in similar fashions to Schmidt et al. 2028, I would argue that additional references and discussion are worth adding to the manuscript. Using the high reproducibility of their system the implications for in situ studies of bacterial colonisation patterns are immense.

I am surprised that the authors are not mentioning one of the usual major limitations of FISH, which, depending on the fixation protocol works either well on either Gram positive or Gram negative cells. Since the authors used a fixation protocol that is usually suitable for Gram negative cells, but render Gram positive cells impermeable to FISH probes, how sure are the authors that the technology will work for Gram positives as well? In their 30 strain synthetic community, Gram positives are underrepresented and are constituted of only Actinobacteria, Firmicutes are missing. I would like to see the authors commenting on this in the discussion and possibly offer solutions to how to use their methods on Gram + bacteria in situ.

In the discussion, the authors focus on situ single-cell transcriptomics an extremely exciting prospect. However arguably this would be initially limited to individual targets. What I am completely missing is the aspect of understanding spatial patterning within bacteria communities - are bacterial taxa closer to the gut lumen vs. the inside of the gut? Are some genera closer to the epidermal ridges in the rhizosphere? Do some taxa spatially associate with each other? Check the papers of the group of Holger Daims in the institute of Michael Wagner, and others.

The discussion is not mentioning the results of the root observation at all - some of the results around the plant root should be moved into the discussion. It is also unclear to me how the authors determined that most of the community was clustered - how was that determined?

What would happen if the authors would use their setup on an environmental sample with unknown bacteria. Would it be possible to use probes with semi-randomised sequences? Based on those hybridisation patterns, would it be possible to predict their identity afterwards?

In the supplements, the authors explain their image analysis pipeline and explain that Phase contrast images are used. This works for clean environments, but not for plant roots. Hence, on plant roots, a different total biomass reference is needed. I understand that the authors are using a general bacterial probe. The signal of this general probe will however strongly depend on the number of ribosomes in the cells, either depending on the metabolic activity of the cells or based on the species. The differences in signal will lead to a difference in signal, which causes issues for thresholding. How did the authors get around these issues? Would a general DNA dye be a better solution to this issue?

Line 26, With all due respect to the number of studies that the authors list to introduce the spatial

organisation in different environments, this has also been performed in the phyllosphere using FISH. In 2014, a team around Julia Vorholt was able to use FISH to determine inter-taxon spatial relationships to each other.

Materials and methods:

I feel that the description of the imaging and mounting of plant roots deserves more details, it is unclear to me how it was mounted since the roots will be a lot thicker than the synthetic mixtures of bacteria. Did you use special spacers? I can imagine that using a NA1.5 objective will lead to an extremely short working distance, was the working distance long enough to image the side of the roots? Did you use a different objective for these images?

During the Multi-round fish procedures, how were buffers changed? was this an automatic process? Were the chambers dry in between (i can imagine that air bubbles would be terrible to get rid of then) or was there a gradient shift of one buffer into the next one?

When going through your workflow to analyse images, there doesn't seem to be a distinction between cultures and roots. I don't understand how this is possible - Phase contrast will not work on the root sample and hence, you are lacking a step in your workflow. Please describe the workflow for roots individually or add the differences into the protocol if they could be described as "either / or"

General question: what is the chance of loosing cells in the incubation chamber during heating and washing - the data should be present already and probably should be shown as well. If there was any and I can imagine that especially on roots, there will be, how did you account for this in your analysis.

Is the chance of loosing cells different if roots are under investigation? in the context of samples where phase contrast is not possible, would it be prudent to maintain a constant dye / or probe in the system in lieu of a phase contrast image?

Supplements - Last line on page two mentions that "Images are analyzed by custom MATLAB scripts." These scripts need to be made available to the community. Otherwise, the methods are not reproducible for the community.

I find the methods used in the supplements fascinating since they themselves would be worth a protocol paper, given that the body of the main is not too long and not overburdened by figures I would even like to see most of the supplements to be integrated into the main text. It is my understanding that the main text can be up to 5000 words long and that the material and methods does not count towards the 5000 words. The authors have another 3000 words available for material and methods - this will give them some space to add more beef to the bone of the manuscript.

This is a minor point for this study, but I am wondering in how far the image analysis holds if the cell density is higher and possibly multi species biofilms are investigated. I think the authors may want to discuss this and possibly give possible solutions this limitation or at least acknowledge it.

This is the first time that I see that the watershed of a phase contrast image is used to segment a thresholded image. Very elegant, but there are potential limitations to very clean background that are free of additional high contrast particles, so the application of this technique in situ will be limited to samples that provide a very clean background.

Figure S9 Panel A, do you think that the probe PD1 is inefficient in labelling the strains on the roots?

Reviewer #2:

Remarks to the Author:

The manuscript by Cao et al. reports the development and application of a highly multiplexed FISH

method for investigating micron-scale spatial organization in microbial communities. Multiplexing is achieved by multiple rounds of hybridization, imaging, and dissociation of the probes. Application of the method is demonstrated by inoculating a 12-taxon model community onto Arabidopsis roots and carrying out imaging one week later.

This is a well-written, technically strong paper. The primary innovation is the use of very short hybridization times (3 minute incubation, total time ~15 minutes per round of hybridization) which enables multiple successive rounds of hybridization. These short hybridization and dissociation times are enabled by the use of microfluidics in a way similar to that reported in a very recent paper by Daniel Dar et al. (published after submission of this manuscript, *Science* 373: 758). The paper by Cao et al. is also commendable for the attention it pays to non-specific binding and the need to design robust strategies for designing combinatorial labels to mitigate the effects of possible off-target binding of rRNA-targeted probes.

The weakness of the manuscript is that it over-estimates the gain in multiplicity that can be achieved with this "SEER-FISH" method compared to previously published methods. Figure 1D, supplementary table 1, and the text lines 96-98 all compare the realized multiplicity of CLASI-FISH (120 distinct labels) and HiPR-FISH (1023 distinct labels) not with the realized multiplicity of SEER-FISH (30 distinct labels) but with its theoretical multiplicity of 10^5 or more. Both CLASI-FISH and HiPR-FISH also have a theoretical multiplicity of nearly 10^5 , depending on the number of fluorophores used. CLASI-FISH creates combinatorial labels by simultaneous hybridization with combinations of up to 16 fluorophores; HiPR-FISH uses a two-step hybridization process, with specificity generated by unlabeled probes which are then detected by a readout hybridization with fluorophore-labeled probes. Both CLASI-FISH and HiPR-FISH can detect $2^N - 1$ distinct targets, where N is the number of fluorophores used, and could therefore, in principle, detect 1023 distinct targets using 10 fluorophores, and 65535 distinct targets using 16 fluorophores. The manuscript by Cao et al. thus does not demonstrate that SEER-FISH represents, in practice, a significant advance in the number of targets that can be discriminated; it rather provides a different way to achieve this multiplicity, using microfluidics rather than spectral imaging. Figure 1D and supplementary table 1 should be modified to compare the methods in an unbiased way, and claims that SEER-FISH is an "unprecedented method" (line 22) with "unparalleled multiplexity" (line 201) should be moderated.

Reviewer #3:

Remarks to the Author:

This paper presents a taxonomy strategy to spatially resolve microbial populations. The presented error correction strategy assigns unique microbial species using R (round), Hamming distance (HD), and samples (S).

The presented technology is a synthesis of published seqFISH and MERFISH methods for the microbial mapping applications.

- 1) The multiplexing, probe design, and microfluidics labeling are the same as in seqFISH, where F^N species can be detected.
- 2) The HD error correction was borrowed from MERFISH, where the error-robust approach refers to the HD calculations of binary barcodes. This HD approach was implemented in seqFISH labeling, increasing the targets.
- 3) The sample (S) addition to the barcoding was also recently named as par-seqFISH (*Science*, Vol. 373, Issue 6556, eabi4882 DOI: 10.1126/science.abi4882), in which 16S sequences were targeted to distinguish bacterial species. Overall, this SEER-FISH is in fact HD implemented seqFISH. As is, it is a useful integration for applicability in microbial mapping. HD's original MERFISH implementation (*Science* 24 Apr 2015: Vol. 348, Issue 6233, aaa6090 DOI: 10.1126/science.aaa6090) was not clearly cited and explained in the paper. Only mentioned 1 to 0 errors but drop outs are common even in color barcodes presented here.
- 4) While the Arabidopsis roots exhibit interesting bacterial populations - the validations are missing. Authors should demonstrate that these detected bacterial species correspond to gold-standard staining to benchmark their studies.
- 5) Multiple roots can be analyzed to visualize root-to-root variations of microbial species.

Similarities and differences can be quantified using SEER-FISH.

6) Supplementary Figs 6 and 8 are noisy - not sure if it can visually distinguish those pure bacterial targets. authors should refine their approach.

7) Fig2G precision and recalls are as low as 0.5 and 0.1 respectively. This agrees well with noisy heatmaps (Fig.2E). There is most likely an issue of probe design or labeling efficiency to distinguish individual bacteria with a cleaner signal.

8) comparing this paper to the par-seqFISH (Science, Vol. 373, Issue 6556, eabi4882 DOI: 10.1126/science.abi4882) is important. Authors should use UMAP analysis to demonstrate how each bacteria is separated in a dimension reduced map.

Overall - this paper can be applicable to microbial imaging based on previously established error correction and multiplexing methods. The application of bacteria in root diversity can be applicable to the plant biology. It could be also interesting to show plant's health and disease in the context of root's bacterial compositions.

Reviewer #1 (Remarks to the Author):

I want to congratulate the authors on this wonderful method and its development. I am fully supportive of the publication of this manuscript albeit i still have minor comments and suggestion for the authors.

Response: We deeply appreciate the reviewer for the acknowledgement of our work and the constructive comments on how to improve the manuscript.

The authors have devised a way that allows them to rapidly hybridise bacteria up to three FISH probes at a time and repeat this process “n” times. Together with an automatic microscopy setup, this allows the error-robust identification of bacterial taxa.

The technique demonstrated in the manuscript holds tremendous potential. The study is barely scratching the surface of the potential utilisation of the method. I feel that the title is slightly overselling the spatial aspect of the study. Although the authors have used their method to profile the distribution of synthetic communities across the length of the tips of an Arabidopsis root, which is truly awesome. Their technique and actually provided images provide an already more detailed insight into the distribution of individual cells on Arabidopsis roots - the spatial relationship between different biomasses at the micrometer resolution can be investigated. Similar approaches albeit less in depths have not been attempted by many, the authors are citing only one of these studies and make no attempt to analyse their data in similar fashions to Schmidt et al. 2028, I would argue that additional references and discussion are worth adding to the manuscript. Using the high reproducibility of their system the implications for in situ studies of bacterial colonisation patterns are immense.

Response: We thank the reviewer for the acknowledgement of potential applications of our method in studying spatial distribution of microbial communities in situ at single cell resolution. We have included additional references and discussion on previous studies of spatial distribution of root microbiome.

Line 169: Previous studies have shown that the composition and spatial distribution of root microbiome is associated with plant physiology and development (Sasse, J. et al, *Trends Plant Sci*, 2018; Chaparo, J.M. et al, *ISME J*, 2014). Rhizosphere microbiome were found to vary across different root types (eg. primary and secondary roots) and regions (Saleem, M. et al, *Microb Ecol*, 2016; Kawasaki, A. et al, *PLoS One*, 2016). Imaging-based approaches have further confirmed the spatial variation in root-colonized microbes (Schmidt, H. et al, *Microbiol Ecol*, 2014; Schmidt, H. et al, *Frontiers in Environmental Science*, 2018; Tovi, N. et al, *Fron Microbiol*, 2019; Massalha, H., et al, *PNAS*, 2019).

I am surprised that the authors are not mentioning one of the usual major limitations of FISH, which, depending on the fixation protocol works either well on either Gram positive or Gram negative cells. Since the authors used a fixation protocol that is usually suitable for Gram negative cells, but render Gram positive cells impermeable to FISH probes, how sure are the authors that the technology will work for Gram positives as well? In their 30 strain synthetic community, Gram positives are underrepresented and are constituted of only Actinobacteria, Firmicutes are missing. I would like to see the authors commenting on this in the discussion and possibly offer solutions to how to use their methods on Gram + bacteria in situ.

Response: We agree with the reviewer that the fixation protocol is critical for FISH. The fixation protocol used in our study is designed to permeabilize both Gram- and Gram+ cells. Because PFA is impenetrable for the thick cell wall of Gram+ bacteria, Gram+ bacteria are recommended to be fixed with 50% ethanol (<https://www.arb-silva.de/fish-probes/fish-protocols/>). Thus we fixed our samples with both 4% PFA and 50% ethanol before hybridization. This fixation protocol is generally used for soil microbiome samples (Ref 7, 47). The description of fixation in the methods section was too brief and may have caused this confusion, so we have revised the description accordingly. Moreover, we have added comments in discussion about how to improve the fixation of Gram-positive bacteria.

Line 270: For fixation, cells were resuspended and incubated in 4% paraformaldehyde (DF0135-2; Leagene) at 4°C for 3 hours, followed by washing with PBS and resuspension in 50% ethanol (for permeabilization of gram-positive cells) (Roller, C. et al, *Microbiology*, 1994). Then cells in 50% ethanol were stored at -20°C before FISH imaging.

Line 242: Depending on the samples, the workflow of SEER-FISH can be improved in several aspects. Firstly, to improve the hybridization efficiency of FISH probes, samples can be pretreated with lysozyme to increase the permeability of cells (especially for gram-positive bacteria) and/or incubation with high-concentration paraformaldehyde for thorough denaturation of rRNA.

The 30-strain synthetic community consists of four phyla: *Proteobacteria*, *Actinobacteria*, *Firmicutes*, and *Bacteroidetes*. One strain abbreviated as PA belongs to *Firmicutes*. We have clarified the strain information of Gram staining and taxonomy in Supplementary Table 2.

In the discussion, the authors focus on situ single-cell transcriptomics an extremely exciting prospect. However arguably this would be initially limited to individual targets. What I am completely missing is the aspect of understanding spatial patterning within bacteria communities - are bacterial taxa closer to the gut lumen vs. the inside of the gut? Are some genera closer to the epidermal ridges in the rhizosphere? Do some taxa spatially associate with each other?

Check the papers of the group of Holger Daims in the institute of Michael Wagner, and others.

Response: We thank the reviewer for the suggestions. We have expanded the discussion on the prospect of mapping biogeography of microbiome and cited the relevant references.

Line 249: The proof-of-principle demonstration of SEER-FISH on plant samples has revealed the heterogeneity in root colonization of microbial communities. We envision that the application of multiplexed FISH methods will greatly facilitate the understanding of biogeography of host-associated bacteria communities at single-cell resolution. Are some microbial taxa enriched in specific regions of plant rhizosphere and animal gut? What is the spatial association among different microbial taxa? The differential distribution of microbial communities on the *Arabidopsis* rhizoplane may be linked to localized immune responses or secreted exudates of plant roots. Moreover, alterations in the spatial structure of microbiome during host development, stress response (biotic and abiotic) (Berry, D. et al., *PNAS*, 2013;

Riva, A. et al., *Nat Commun*, 2019) and diseases could lead to novel insights in host-microbiome interactions.

The discussion is not mentioning the results of the root observation at all - some of the results around the plant root should be moved into the discussion. It is also unclear to me how the authors determined that most of the community was clustered - how was that determined?

Response: We thank the reviewer for the constructive comments. Following the suggestions (as well as comments from other reviewers), we have expanded the analysis and discussion on the plant roots. In particular, we have included quantitative analysis on the coaggregation of bacterial cells using the linear dipole algorithm (Daims, H. & Wagner, *Methods Enzymol*, 2011; Remus-Emsermann, M. N. et al, *Environ Microbiol*,2014), shown in Figure 5E and Supplementary Figure 12.

Line 190: We further analyzed the coaggregation of bacteria cells using the linear dipole algorithm (Daims, H. & Wagner, *Methods Enzymol*, 2011; Remus-Emsermann, M. N. et al, *Environ Microbiol*,2014) (**Fig. 5C-E, see Methods**). Indeed, the pair correlation function revealed that microbes on the root surface coaggregated at distance up to ~30 μm , which was consistent across 3 root samples and in agreement with visual inspections (**Fig. 4 and 5C**).

Fig. 5E. The spatial correlation of root-colonized bacterial cells is analyzed by linear dipole algorithm (Methods). The solid lines indicate the pair correlation between bacterial cells, the shadows indicate the 95% confidence intervals estimated by sampling different regions on each root. The horizontal dash line ($g(r)=1$) refers to the expected value of a randomized spatial distribution.

Line 414: Spatial analysis of root-colonized microbial communities

The coaggregation of bacterial cells is analyzed by the linear dipole algorithm as previously described. (Daims, H. & Wagner, *Methods Enzymol*, 2011; Remus-Emsermann, M. N. et al, *Environ Microbiol*,2014). Briefly, the pair cross-correlation function between two bacterial populations is defined as

$$g(r) = \frac{P_r}{2D_i D_j}$$

where P_r is the probability for a dipole with length r hit the bacteria from the two populations. D_i indicated the density of population i . To analyze bacterial cells belonging to the same population, the correlation function is defined as

$$g(r) = \frac{P_r}{D^2}$$

where D is the density of the population of interest. If bacterial cells are randomly distributed at distance r , $g(r) = 1$; in contrast, if bacterial cells coaggregate at distance r , $g(r) > 1$.

For each image, a mask of the root is manually drawn as the region of interest based on the phase-contrast image. Linear dipoles are evaluated only if both ends fall into the masked regions, therefore, bacterial cells not on the root surface are excluded from the analysis. The cell density is calculated as the area cover by the bacteria cells divided by the area of the root.

What would happen if the authors would use their setup on an environmental sample with unknown bacteria. Would it be a possible to use probes with semi-randomised sequences? Based on those hybridisation patterns, would it be possible to predict their identity afterwards?
Response: To label microbiome samples, we currently use rRNA sequences obtained from sequencing to guide probe design. The reviewer raised an interesting proposal to identify unknown bacteria using probes with randomized sequences. In principle, with clever design of FISH probes, this could possibly turn the imaging approach into *in situ* microbiome sequencing. We thank the reviewer for the interesting idea, which we may explore in future studies.

In the supplements, the authors explain their image analysis pipeline and explain that Phase contrast images are used. This works for clean environments, but not for plant roots. Hence, on plant roots, a different total biomass reference is needed. I understand that the authors are using a general bacterial probe. The signal of this general probe will however strongly depend on the number of ribosomes in the cells, either depending on the metabolic activity of the cells or based on the species. The differences in signal will lead to a difference in signal, which causes issues for thresholding. How did the authors get around these issues? Would a general DNA dye be a better solution to this issue?

Response: In our image analysis pipeline, we used the cross-correlation between phase contrast images for the alignment of multi-round imaging, which worked well for root samples. For cell segmentation of root samples, we used the inverted fluorescence images of the universal probe EUB338 (see panel C of Supplementary Fig. 2). Although the variations in the fluorescence signal of the universal probe did not cause any issue for thresholding in our samples, we agree with the reviewer that a general DNA dye may provide a relatively uniform signal and thus a better solution.

Line 242: Depending on the samples, the workflow of SEER-FISH can be improved in several aspects. ... Thirdly, a general nucleic acid dye could be incorporated in multi-round imaging to

label bacterial cells with a uniform signal (Schmidt, H. et al., *Frontiers in Environmental Science*, 2018).

Line 26, With all due respect to the number of studies that the authors list to introduce the spatial organisation in different environments, this has also been performed in the phyllosphere using FISH. In 2014, a team around Julia Vorholt was able to use FISH to determine inter-taxon spatial relationships to each other.

Response: We thank the reviewer for providing a useful reference. We have included this reference with respect to the introduction of spatial structure of microbial communities (line 26).

Materials and methods:

I feel that the description of the imaging and mounting of plant roots deserves more details, it is unclear to me how it was mounted since the roots will be a lot thicker than the synthetic mixtures of bacteria. Did you use special spacers? I can imagine that using a NA1.5 objective will lead to an extremely short working distance, was the working distance long enough to image the side of the roots? Did you use a different objective for these images?

Response: We thank the reviewer for the questions that help us clarify the methods. We used a silicone spacer between the coverslip and the slide. We used a NA1.45 objective in imaging (apologies for the typo in Methods). The working distance of the objective is 0.13 mm, while the roots radius is 50-100 μm . Therefore, we could image the side of roots using the objective. We have added details in Methods and Fig. S1B to describe the imaging and mounting of samples.

Line 315: Fixed samples were first adhered onto adhesive coverslips (40 mm round, 0.15 mm thick); then a silicone gasket (40 mm round, 0.75 mm thick) with a central rectangle cavity was placed on the coverslip; and then the micro-aqueduct slide was placed on the gasket. The coverslip, gasket and micro-aqueduct slide constitute a sandwich structure. Samples were in the cavity where buffers passed through.

Fig. S1B. Schematic diagram of SEER-FISH experimental setup. A flow chamber (Biotech FCS2) is secured into a stage adapter to interface with a microscope for imaging. Silicone gasket (40 mm round, 0.75 mm thick) with a rectangle cavity internal that separates the micro-aqueduct slide from the coverslip (40 mm round, 0.15 mm thick) is used to create an optical cavity in the chamber. Laminar flow perfusion that comes into one of the ports (INLET) on one side of the chamber is collected within the optical cavity and then directed out of the chamber on the other side (OUTLET). Uniform temperature across the entire field is maintained by a

temperature controller. Flow through this chamber is controlled via an extraneous peristaltic pump.

Line 328: a Plan Apo λ 100 x oil objective lens (Nikon, 1.45 NA)

During the Multi-round fish procedures, how were buffers changed? was this an automatic process? Were the chambers dry in between (i can imagine that air bubbles would be terrible to get rid of then) or was there a gradient shift of one buffer into the next one?

Response: We have added more details in the methods section to clarify these points. Chamber was not dry between changes in buffer, and FCS2 chamber was designed to allow high-volume laminar flow perfusion avoiding a gradient shift. Currently the process is manually controlled; the fully automatic process still requires some technical optimization (e.g. flow rate), so we didn't include in this manuscript.

Line 319: Fluidics was controlled via a peristaltic pump (LongerPump, BT100-2J) and set at a constant flow velocity of 500 μ l/min (10 rpm). Buffers were warmed by a metal bath at 50°C to get rid of dissolved oxygen before passing through the flow chamber.

When going through your workflow to analyse images, there doesn't seem to be a distinction between cultures and roots. I don't understand how this is possible - Phase contrast will not work on the root sample and hence, you are lacking a step in your workflow. Please describe the workflow for roots individually or add the differences into the protocol if they could be described as "either / or"

Response: We thank the reviewer for helping us to clarify the methods. As discussed above, for root samples, we used the phase contrast images for multi-round alignment and the inverted fluorescence images of the universal probe EUB338 for cell segmentation (Supplementary Fig. 2C). The workflow was described in Supplementary Figure 2 and Methods.

Line 339: Then, the phase contrast image (for *in vitro* bacterial communities) or inverted fluorescence image with the universal probe EUB338 (for root-associated bacterial communities) were segmented into binary mask images using an adaptation threshold followed by the watershed algorithm (Supplementary Fig. 2).

Fig. S2C. For *in vivo* microbiome samples, a fluorescent image is acquired by labeling bacteria with the universal probe EUB338, and the inverted image is used for segmentation as shown in panel B.

General question: what is the chance of losing cells in the incubation chamber during heating

and washing - the data should be present already and probably should be shown as well. If there was any and I can imagine that especially on roots, there will be, how did you account for this in your analysis.

Response: We calculated the proportion of losing cells for *in vitro* cultures and roots. Roughly, for *in vitro* cultures, 3% of bacterial cells were lost after 26 rounds of imaging (indicated in line 88-89); for root samples, ~15% of bacterial cells were lost after 8 rounds of imaging (indicated in line 180-181). The fraction of lost cells for root samples are shown in Supplementary Figure 10. We used a strict criterion to estimate the cells lost in multi-round imaging, i.e. no fluorescence signal for more than three rounds (indicated in line 346-347); these cells were not included in downstream analysis.

Fig. S10. The fraction of lost and unidentified bacterial cells in multi-round FISH imaging of *Arabidopsis* roots. The fraction of identified, unidentified and lost bacterial cells on 3 roots. The number of bacterial cells detected by the universal probe EUB338 is indicated on top. A bacterial cell is labeled as “lost” if it is not detected in any fluorescence channel for more than 3 rounds.

Line 346: A bacterial cell is labeled as “lost” if it is not detected in any fluorescence channel for more than 3 rounds (**Supplementary Fig. 10**).

Is the chance of losing cells different if roots are under investigation? in the context of samples where phase contrast is not possible, would it be prudent to maintain a constant dye / or probe in the system in lieu of a phase contrast image?

Response: As discussed in the response above, we found that the fraction of losing cells was higher in root samples. For root samples, instead of the phase contrast image, we used the fluorescence of universal probe EUB338 labeling for segmentation, which worked reasonably well in our experiments (Fig. S2C). A constant dye would be nice to track bacterial cells during multi-round imaging, yet misidentification may occur due to non-specific binding to DNA of host cells. We appreciate the reviewer’s suggestion and will try to incorporate dyes (e.g. SYBR Green) in future experiments. We have added this suggestion in discussion about the improvement of SEER-FISH (Line 242).

Supplements - Last line on page two mentions that “Images are analyzed by custom MATLAB scripts.” These scripts need to be made available to the community. Otherwise, the methods are not reproducible for the community.

Response: We have made all custom scripts available to the community.

Line 431: All codes are available on Github: <https://github.com/JacobZuo/SEER-FISH>. The probe design pipeline uses several third-party softwares as described in Methods.

I find the methods used in the supplements fascinating since they themselves would be worth a protocol paper, given that the body of the main is not too long and not overburdened by figures I would even like to see most of the supplements to be integrated into the main text. It is my understanding that the main text can be up to 5000 words long and that the material and methods does not count towards the 5000 words. The authors have another 3000 words available for material and methods - this will give them some space to add more beef to the bone of the manuscript.

Response: After careful consideration of the pros and cons, we decided to keep the manuscript concise and leave the details in the supplement. Regardless, we very much appreciate the reviewer’s recognition of our presentations in the supplement.

This is a minor point for this study, but I am wondering in how far the image analysis holds if the cell density is higher and possibly multi species biofilms are investigated. I think the authors may want to discuss this and possibly give possible solutions this limitation or at least acknowledge it.

Response: We are not sure how our image analysis would perform for very dense bacteria communities such as biofilms. Following the reviewer’s suggestion, we have acknowledged this limitation in the discussion.

Line 242: Depending on the samples, the workflow of SEER-FISH can be improved in several aspects. ... Secondly, the segmentation algorithm that we used is potentially limited to samples with a clean background. Imaging more complicated samples or dense microbial communities (e.g. biofilms) may require improvements in the image analysis workflow.

This is the first time that I see that the watershed of a phase contrast image is used to segment a thresholded image. Very elegant, but there are potential limitations to very clean background that are free of additional high contrast particles, so the application of this technique in situ will be limited to samples that provide a very clean background.

Response: We thank the reviewer for pointing out both the novelty and limitation of our image analysis. As discussed in the response above, we have added discussion on the image segmentation algorithm.

Figure S9 Panel A, do you think that the probe PD1 is inefficient in labelling the strains on the roots?

Response: Although the probe PD1 works reasonably well in labeling the strain in vitro, it is possible that the labeling is inefficient for root samples. This is one plausible explanation for

the observed discrepancy between the relative abundance measured by sequencing and imaging of PD1. In addition, there are several other factors: 1) The recall rate of PD1 is relatively low (Figure 2D). 2) The species composition by 16S sequencing is based on the whole root, while for imaging we only sampled several regions along the root.

Reviewer #2 (Remarks to the Author):

The manuscript by Cao et al. reports the development and application of a highly multiplexed FISH method for investigating micron-scale spatial organization in microbial communities. Multiplexing is achieved by multiple rounds of hybridization, imaging, and dissociation of the probes. Application of the method is demonstrated by inoculating a 12-taxon model community onto Arabidopsis roots and carrying out imaging one week later.

This is a well-written, technically strong paper. The primary innovation is the use of very short hybridization times (3 minute incubation, total time ~15 minutes per round of hybridization) which enables multiple successive rounds of hybridization. These short hybridization and dissociation times are enabled by the use of microfluidics in a way similar to that reported in a very recent paper by Daniel Dar et al. (published after submission of this manuscript, Science 373: 758). The paper by Cao et al. is also commendable for the attention it pays to non-specific binding and the need to design robust strategies for designing combinatorial labels to mitigate the effects of possible off-target binding of rRNA-targeted probes.

Response: We appreciate the reviewer for acknowledging the novelty of our work, and for highlighting our design of error-robust strategies in rRNA-targeted FISH. The paper provided by the reviewer has been included in the reference (please see more discussion on par-seqFISH in our response to reviewer #3).

The weakness of the manuscript is that it over-estimates the gain in multiplicity that can be achieved with this “SEER-FISH” method compared to previously published methods. Figure 1D, supplementary table 1, and the text lines 96-98 all compare the realized multiplicity of CLASI-FISH (120 distinct labels) and HiPR-FISH (1023 distinct labels) not with the realized multiplicity of SEER-FISH (30 distinct labels) but with its theoretical multiplicity of 10^5 or more. Both CLASI-FISH and HiPR-FISH also have a theoretical multiplicity of nearly 10^5 , depending on the number of fluorophores used. CLASI-FISH creates combinatorial labels by simultaneous hybridization with combinations of up to 16 fluorophores; HiPR-FISH uses a two-step hybridization process, with specificity generated by unlabeled probes which are then detected by a readout hybridization with fluorophore-labeled probes. Both CLASI-FISH and HiPR-FISH can detect $2^N - 1$ distinct targets, where N is the number of fluorophores used, and could therefore, in principle, detect 1023 distinct targets using 10 fluorophores, and 65535 distinct targets using 16 fluorophores. The manuscript by Cao et al. thus does not demonstrate that SEER-FISH represents, in practice, a significant advance in the number of targets that can be discriminated; it rather provides a different way to achieve this multiplicity, using microfluidics rather than spectral imaging. Figure 1D and supplementary table 1 should be modified to compare the methods in an unbiased way, and claims that SEER-FISH is an “unprecedented method” (line 22) with “unparalleled multiplexity” (line 201) should be moderated.

Response: We agree with the reviewer that the theoretical multiplicity of CLASI-FISH and HiPR-FISH is not in our comparison of different methods. We have removed the comparison from Figure 1D and provided an unbiased comparison in Supplementary Table 1. The texts have also been modified accordingly.

Line 22: “an unprecedented method” ->“a novel method”

Line 95-97: The multiplexity of SEER-FISH can easily reach 10^5 (**Fig. 1D**), which is comparable to the theoretical multiplexity of existing methods for imaging microbiome samples, such as CLASI-FISH and HiPR-FISH.

Line 205: “unparalleled multiplexity” -> “superior multiplexity”

Reviewer #3 (Remarks to the Author):

This paper presents a taxonomy strategy to spatially resolve microbial populations. The presented error correction strategy assigns unique microbial species using R (round), Hamming distance (HD), and samples (S).

The presented technology is a synthesis of published seqFISH and MERFISH methods for the microbial mapping applications.

1) The multiplexing, probe design, and microfluidics labeling are the same as in seqFISH, where F^N species can be detected.

2) The HD error correction was borrowed from MERFISH, where the error-robust approach refers to the HD calculations of binary barcodes. This HD approach was implemented in seqFISH labeling, increasing the targets.

3) The sample (S) addition to the barcoding was also recently named as par-seqFISH (*Science*, Vol. 373, Issue 6556, eabi4882 DOI: 10.1126/science.abi4882), in which 16S sequences were targeted to distinguish bacterial species.

Overall, this SEER-FISH is in fact HD implemented seqFISH. As is, it is a useful integration for applicability in microbial mapping. HD's original MERFISH implementation (*Science* 24 Apr 2015: Vol. 348, Issue 6233, aaa6090 DOI: 10.1126/science.aaa6090) was not clearly cited and explained in the paper. Only mentioned 1 to 0 errors but drop outs are common even in color barcodes presented here.

Response: We thank the reviewer for acknowledging the value of our methods in mapping microbiome. We have expanded the explanation of MERFISH and clearly indicated its originality in HD error correction strategies. We have also included discussion on the par-seqFISH paper (please see the response to point 8 below).

Line 228: The incorporation of error-correction strategies, originally implemented in MERFISH for multi-round mRNA profiling (Chen, K.H. et al., *Science*, 2015; Moffitt, J.R. et al., *PNAS*, 2016), is expected to improve the accuracy of target identification, but has not been studied in the context of microbiome. By incorporating error-robust encoding schemes in SEER-FISH, we show that the precision and recall of taxonomic identification can be improved, particularly in scenarios where non-specific hybridization is unavoidable. In mRNA labeling, probe specificity is not a major concern and sparse binary HD codes were used in MERFISH to mitigate "dropout" errors (1→0 in barcodes). In contrast, detection errors in bacterial rRNA FISH are mainly caused by non-specific (i.e., off-target) labeling of phylogenetically related rRNA sequences. In our study, we chose to exclude the non-fluorescent code in the codebook (i.e., color code=0) to minimize detection errors caused by non-specific labeling (color code $F \rightarrow F'$).

4) While the Arabidopsis roots exhibit interesting bacterial populations - the validations are missing. Authors should demonstrate that these detected bacterial species correspond to gold-standard staining to benchmark their studies.

Response: Following the reviewer's suggestion, we have included imaging experiments that validate the bacterial cells detected on roots by DAPI staining, the universal probe EUB338 and the negative control probe NON338 (complementary to EUB338). Images of both axenic (i.e. germ-free) roots and bacteria-inoculated roots are shown in Fig. S9. The co-localization of

EUB338 fluorescence signal and DAPI staining in the bacteria-inoculated root sample indicates that bacterial cells on Arabidopsis roots are correctly identified by FISH.

Line 177: Root-colonized bacterial cells were detected by the universal FISH probe EUB338, and further validated by DAPI staining and the negative control FISH probe NON338 (Supplementary Fig. 9).

Fig. S9. Validation of root-colonized bacteria by DAPI staining. Images of root samples (axenic and bacteria-inoculated) labeled by DAPI and FISH (the universal probe EUB338 and the negative control probe NON338). The co-localization of EUB338 fluorescence signal and DAPI staining indicates that bacterial cells on Arabidopsis roots are correctly identified by FISH. Scale bars, 50 μm .

5) Multiple roots can be analyzed to visualize root-to-root variations of microbial species. Similarities and differences can be quantified using SEER-FISH.

Response: We thank the reviewer for the constructive comments. Following this suggestion, we have performed SEER-FISH on multiple roots and analyzed the similarities and differences in root colonized microbial communities. The community composition and the clustering of bacterial cells were consistent across three root samples, while the coaggregation of the two most abundant species AD1 and AG1 showed root-to-root variations. The results have been included in the manuscript in Figure 4-5 and Supplementary Figure 11-12.

Line 179: We imaged multiple regions on 3 root samples (within ~5mm to the root tip) and quantified the community composition by SEER-FISH (Fig. 4B-D). Roughly 15% of bacterial cells were lost after eight rounds of imaging (Supplementary Fig. 10). In the regions that we imaged on 3 root samples, a total of ~15000 bacterial cells (of 12 species) were successfully identified (including error corrections), and only ~10% cells were unidentified. While we observed variations across different regions (Supplementary Fig. 11), the overall community composition estimated by SEER-FISH were highly similar across 3 roots (Pearson correlation $R > 0.97$, $P < 10^{-5}$). There was also close agreement between the community composition estimated by SEER-FISH and by 16S rRNA amplicon sequencing of root samples (Fig. 4E).

Line 190: We further analyzed the coaggregation of bacteria cells using the linear dipole algorithm (Fig. 5C-E, see Methods). Indeed, the pair correlation function revealed that microbes on the root surface coaggregated at distance up to $\sim 30 \mu\text{m}$, which was consistent across 3 root samples and in agreement with visual inspections (Fig. 4 and 5C). Furthermore, we analyzed the pair cross-correlation of the two most abundant species AD1 and AG1 (Supplementary Fig. 12). We found coaggregation of AD1 and AG1 on root 3, nevertheless, there was substantial variations among the regions that we imaged. The root-to-root variations in spatial patterns suggest stochasticity in bacteria colonization, which should be taken into account in future studies.

Figure 4. Spatial profiling of microbial communities colonized on Arabidopsis roots by SEER-FISH. **A)** The protocol of synthetic microbial community colonization on Arabidopsis roots. Arabidopsis seeds were germinated on an MS plate and then colonized by a synthetic community of 12 bacterial species for 7 days under hydroponic conditions (Methods). **B-D)** The colonization of 12-species synthetic community on different regions of 3 independent roots as identified by SEER-FISH. The edges of the root (white lines) are drawn manually based on the phase-contrast image. Scale bars, 100 μm . Numbers below each image indicate the distance to the root tip. **E)** The composition of the communities colonized on root measured by 16S amplicon sequencing (mixture of 10 roots) and SEER-FISH (the Pearson correlation is indicated by orange dash lines). The Pearson correlation between the community compositions on 3 roots measured by SEER-FISH is indicated by blue lines.

Fig. S12. Spatial analysis of microbial communities on Arabidopsis roots. The coaggregation of bacterial cells is analyzed by the linear dipole algorithm (see Methods). The solid lines indicate the pair cross-correlation between bacterial cells, the shadows indicate the 95% confidence intervals estimated by sampling different regions on each root. The horizontal dash line ($g(r)=1$) refers to the expected value of a randomized spatial distribution.

6) Supplementary Figs 6 and 8 are noisy - not sure if it can visually distinguish those pure bacterial targets. authors should refine their approach.

Response: While there is some degree of non-specificity, the fluorescence intensity of specific probe-species pairs (diagonal) is significantly higher than non-specific pairs (off-diagonal) (Fig. 2F). In our probe design pipeline, we required at least 3 mismatches (Fig. S4); a more stringent criteria can be applied, yet would limit the possible targets in rRNA. We agree with the reviewer that future efforts can be made to improve probe specificity and have included this point in the discussion.

Line 240: Other experimental modifications to reduce non-specific hybridization, such as increasing hybridization stringency, adding competitor probes for off-target taxa or dual probes with overlapping specificity, can also be used to improve accuracy and are readily compatible with SEER-FISH.

The real challenge of mapping complex microbial communities is that we often cannot find perfectly “orthogonal” rRNA-targeting FISH probes at species/strain level. To label microbiome samples with dozens to hundreds of species, it is not feasible to perform probe validations on pure culture (as shown in Supplementary Figure 6), and then optimize the probe design via iterations. Based on our experiments and simulations (Figure 2, Supplementary Figure 7-8), we demonstrate that error-robust encoding can tolerate some degree of non-specificity in designed FISH probes and achieve high accuracy in taxonomic identification.

7) Fig2G precision and recalls are as low as 0.5 and 0.1 respectively. This agrees well with noisy heatmaps (Fig.2E). There is most likely an issue of probe design or labeling efficiency to distinguish individual bacteria with a cleaner signal.

Response: In Fig. 2G, the simulation results (based on the experimentally measured probe specificity for 12 species) showed that error-robust encoding schemes ($HD \geq 2$, blue and $HD \geq 4$, red) can substantially improve the accuracy of taxonomic identification, compared with the poor performance for coding schemes that do not tolerate errors ($HD \geq 1$, black, which the reviewer noted). Therefore, we used HD4 error-robust encoding in experiments, and showed that SEER-FISH achieved excellent performance in precision (median=0.98) and recall (median=0.89) (Fig. 2D, Supplementary Fig. 5C).

Similar to the response above, we fully agree with the reviewer that improvement in probe specificity would lead to better precision and recall in taxonomic identification, as we showed by simulations in Supplementary Figure 7.

8) comparing this paper to the par-seqFISH (Science, Vol. 373, Issue 6556, eabi4882 DOI: 10.1126/science.abi4882) is important. Authors should use UMAP analysis to demonstrate how each bacteria is separated in a dimension reduced map.

Response: Following the reviewer's suggestion, we have included analysis on dimension-reduced maps (**Supplementary Figure 13**). The multi-round multi-color imaging data were visualized in a dimension reduced map by t-SNE. For both simulated and real imaging data, we found that bacterial cells of the same species were clustered in the dimension reduced map, and different species were clearly separated. This demonstrates that our encoding-decoding strategy can correctly identify each bacterial taxa. In future studies, the dimensionality reduction approach may be used to identify unknown microbial taxa (related to reviewer #1's proposal of semi-randomized probe sequences). We have included these results and comparison to par-seqFISH in the discussion.

Line 217: Recently, a sequential FISH method reported as par-seqFISH spatially profiled the expression of ~100 marker genes in bacterial populations of *Pseudomonas aeruginosa* at single-cell resolution (Dar, D. et al., *Science*, 2021). par-seqFISH focused on spatial transcriptomics within a bacterial population (of one species), while SEER-FISH was developed to study spatial metagenomics of a multi-species microbial community. In par-seqFISH, mRNAs were labeled once with a nonbarcoded approach (i.e. the multiplexity scales linearly with the number of imaging rounds); while in SEER-FISH, rRNAs were labelled repeatedly with error-robust encoding (i.e. the multiplexity scales exponentially with the number of imaging rounds). Inspired by the dimensionality reduction approaches commonly used in single cell transcriptomics analysis, we visualized the multi-round, multi-color SEER-FISH imaging data in dimension reduced maps by t-SNE (**Supplementary Fig. 13**). For both simulated and real imaging data, we found that bacterial cells of the same species were clustered in the dimension reduced map, and different species were clearly separated. In future studies, the dimensionality reduction approach may be used to identify unknown microbial taxa.

Fig. S13. Dimensionality reduction analysis of the multi-round, multi-color imaging data acquired by SEER-FISH. (A-B) t-SNE analysis of the simulated data of a 12-species synthetic community. The 8-round 3-color codebook is the same as Table S5. The specificity parameters for specific, non-specific and background hybridization is set as -0.3, -0.9 and -2 according to the ΔG , respectively. For each data point (i.e. bacterial cell), the 24 dimensional data (8 imaging rounds \times 3 fluorescence channels) is reduced to 2 dimensions by ‘tsne’ function in MATLAB (parameters are set as, LearnRate: 200; Perplexity: 20; Exaggeration: 10). The points are colored by species known as ground truth (in panel A) or inferred by our encoding-decoding strategy (in panel B). Unidentified cells are labeled as grey dots in (B). (C) t-SNE analysis of the imaging data of 12 species (data from Fig 3A). The points are colored by identified species. Unidentified and lost cells are labeled as grey and black dots, respectively.

Overall - this paper can be applicable to microbial imaging based on previously established error correction and multiplexing methods. The application of bacteria in root diversity can be applicable to the plant biology. It could be also interesting to show plant's health and disease in the context of root's bacterial compositions.

Response: We thank the reviewer for suggesting future applications in plant biology, including the spatial structure of rhizosphere microbiome and its association with plant health. We expanded the discussion accordingly (Line 249-256).

Reviewers' Comments:

Reviewer #1:

Remarks to the Author:

I believe that the authors have addressed all my comments appropriately. Wonderful study.

One last correction. "sp." after genus names should not be italicized

Reviewer #2:

Remarks to the Author:

The authors have satisfactorily addressed my previous concerns. I have some comments on the newly added material:

In the linear dipole analysis (Figure 5E and Figure S12), please make clear whether analysis was of auto-correlation (1 population) or cross-correlation (2 populations). The use of the term "co-aggregation" (line 192, Figure 5E label on plot, and Figure S12 legend) implies aggregation of distinct populations (cross-correlation), but I believe figure 5D shows a one-population analysis of all cells simultaneously and most of the aggregation signal appears to be coming from auto-correlation of *Acidovorax* and auto-correlation of *Agrobacterium* (figure S12).

line 245: How would high-concentration paraformaldehyde cause denaturation of rRNA? Do the authors mean formamide rather than paraformaldehyde?

line 272: It is not correct to cite reference (56) for the use of 50% ethanol to permeabilize gram-positive cells. Reference 56 (Roller et al. 1994) finds that fixation of gram-positive cells in 50% ethanol INSTEAD OF paraformaldehyde improves the fraction of cells reactive with the FISH probe. This is different from the procedure followed by the authors of the current paper, fixation in paraformaldehyde followed by resuspension in 50% ethanol.

line 217: please correct spelling of *Pseudomonas*

Reviewer #3:

Remarks to the Author:

Authors have provided a revision of this seqFISH integrated with error correction for profiling microbial species.

- 1) Regarding the validation experiment (Root-colonized bacterial cells were detected by the universal FISH177 probe EUB338, and further validated by DAPI staining and the negative control FISH probe NON338) - authors should provide quantification of overlapping FISH signal with DAPI to demonstrate that single-cells can be detected in this validation. Also at least 3 replicates must be shown to describe statistical significance of validations.
- 2) Regarding 3 root comparisons, the spatial analysis of microbial neighborhoods are missing. For instance, authors should zoom into regions and quantify single-cells that are located near each other. Spatial distributions across tip length can also be done other techniques even using root cutting + PCR, but seqFISH can map out cellular interactions in local regions of the data. Right now, the comparison data is rather weak. What do we learn from root variations, anything we did not know before?
- 3) Please remove the novel statement from the abstract. The updated manuscript made it clear that this technique is in fact seqFISH + HD from MERFISH, no novelty here. Also multiplexing was also indicated to be similar with CLASI-FISH25 and HiPR-FISH and par-seqFISH. Authors should only focus on the biological findings from the experiments as an application of existing FISH methods that are already used in this platform.

4) For multiplexing, it talks about 10^5 multiplexing, but this paper indicated the use of only 12 to 30 probes (supplementary table 2). Authors should remove highly multiplexed claims from this manuscript without demonstrating that thousands multiplexing is feasible because they are unsure about density of microbes.

5) HD correction in MERFISH is said to be related to drop outs but not non specific binding. In fact, MERFISH or seqFISH error correction (<https://www.ncbi.nlm.nih.gov/pmc/articles/PMC6544023/>) both captures drop out or nonspecific calling. authors should remove this statement as it may not be correct. Any barcode does not correspond to the original assignment will be removed, making the correction regardless of false positives or negatives.

After the revision, it has become apparent that the novelty of barcode correction is no longer relevant here. The biological findings also remain weak because spatial neighborhood analysis or any functional microbial mapping have not been included that can show us something previously unknown.

Thus, the revision dampened the enthusiasm towards this manuscript due to clarification of the technique to be a seqFISH only and the findings did not yield any biological novel insights. Why would you observe some bacteria in a specific location near another bacteria or why are they located in different length ranges.

Also, the revision seems to be rushed with only $n=3$ roots while the biology of roots can rather be complex and biological significance may take more samples. Validation experiments have not even been quantified or statistically measured. Multiplexing is only 30 species that can also be measured by CLASI-FISH25 and HiPR-FISH and 10^5 multiplexing is unsupported because authors do not have even a good 16S classifier for these many species. It remains as a hype in this current version.

REVIEWER COMMENTS

Reviewer #1 (Remarks to the Author):

I believe that the authors have addressed all my comments appropriately. Wonderful study.
One last correction. "sp." after genus names should not be italicized.

Response: We thank the reviewer again for helping us improve the manuscript. We have corrected the font of "sp.".

Reviewer #2 (Remarks to the Author):

The authors have satisfactorily addressed my previous concerns.

Response: We thank the reviewer again for helping us improve the manuscript.

I have some comments on the newly added material:

In the linear dipole analysis (Figure 5E and Figure S12), please make clear whether analysis was of auto-correlation (1 population) or cross-correlation (2 populations). The use of the term "co-aggregation" (line 192, Figure 5E label on plot, and Figure S12 legend) implies aggregation of distinct populations (cross-correlation), but I believe figure 5D shows a one-population analysis of all cells simultaneously and most of the aggregation signal appears to be coming from auto-correlation of *Acidovorax* and auto-correlation of *Agrobacterium* (figure S12).

Response: Following the reviewer's suggestion, we revised Figure 5D and Figure S12 to make clear whether analysis was auto-correlation or cross-correlation. The reviewer is correct that Figure 5D shows a one-population analysis of all cells simultaneously, while Figure S12 shows both auto-correlation and cross-correlation. Also, we changed "co-aggregation" to "cluster" to avoid misunderstanding.

Fig. 5D. The spatial correlation of root-colonized bacterial cells is analyzed by linear dipole algorithm (Methods). The solid lines indicate the auto correlation between bacterial cells, the shadows indicate the 95% confidence intervals estimated by sampling different regions on each root. The horizontal dash line ($g(r)=1$) refers to the expected value of a randomized spatial distribution.

line 245: How would high-concentration paraformaldehyde cause denaturation of rRNA? Do the authors mean formamide rather than paraformaldehyde?

Response: We thank the reviewer for pointing out the typo. We meant "formamide". The texts have been corrected.

line 272: It is not correct to cite reference (56) for the use of 50% ethanol to permeabilize gram-positive cells. Reference 56 (Roller et al. 1994) finds that fixation of gram-positive cells in 50% ethanol INSTEAD OF paraformaldehyde improves the fraction of cells reactive with the FISH probe. This is different from the procedure followed by the authors of the current paper, fixation in paraformaldehyde followed by resuspension in 50% ethanol.

Response: We thank the reviewer for pointing out the incorrect citation. Our fixation protocol

is adapted from Llobet-Brossa et al. 1998 (Ref 62), which is generally used for soil samples (<https://www.arb-silva.de/fish-probes/fish-protocols/>). We have added the correct citation in Methods section.

line 217: please correct spelling of Pseudomonas

Response: We thank the reviewer for pointing out the typo. We have corrected the spelling of Pseudomonas.

Reviewer #3 (Remarks to the Author):

Authors have provided a revision of this seqFISH integrated with error correction for profiling microbial species.

Response: We thank the reviewer again for helping us improve the manuscript.

1) Regarding the validation experiment (Root-colonized bacterial cells were detected by the universal FISH probe EUB338, and further validated by DAPI staining and the negative control FISH probe NON338) - authors should provide quantification of overlapping FISH signal with DAPI to demonstrate that single-cells can be detected in this validation. Also at least 3 replicates must be shown to describe statistical significance of validations.

Response: Following the reviewer's suggestion, we have provided quantification of overlapping FISH signal with DNA dyes to validate the detection of bacterial cells (Fig. S9). SYBR has been found to have superior signal-to-noise ratio than DAPI for staining rhizoplane-colonizing bacteria (Ref 42 and 47). Thus, we used FISH probe EUB338 and SYBR Safe to image roots colonized by two different bacterial communities (SynCom13, n=3 replicates; SynCom22, n=3 replicates). We found that $97.8 \pm 1.5\%$ EUB338 labeled cells were labeled by SYBR, and $95.5 \pm 2.0\%$ SYBR labeled cells were labeled by EUB338 (Fig. S9A-B). Also, there was no signal for negative control FISH probe NON338 on bacteria-colonized roots. Based on previous experiments, we also quantified the overlapping FISH signal with DAPI, and found that about 90% EUB338 labeled cells were labeled by DAPI (see Figure R1 below, not included in the revised manuscript). Taken together, our results demonstrate that bacteria cells colonized on rhizoplane can be correctly identified.

Fig. S9. Identification of root-colonized bacteria. Root-colonized bacterial cells recognized by universal FISH probe EUB338 were validated by staining with SYBR Safe (Invitrogen). Roots colonized with a 13-species community (SynCom 13) or a 22-species community (SynCom 22) were hybridized with EUB338 probe for 10 min (20% formamide, 46°C), then stained with SYBR Safe (10X) for 15 min at room temperature before imaging. **A.** Representative images of bacteria-colonized root samples labeled by the FISH probe EUB338 and the nucleic acid dye SYBR Safe. Single cells labelled by EUB338 or SYBR Safe are shown below the original images. **B.** Proportion of EUB338 labeled cells that are labeled by SYBR Safe (red bars) and vice versa (green bars) across multiple roots. The total number of cells

imaged for each root is indicated. $97.8\% \pm 1.5\%$ EUB338 labeled cells were labeled by SYBR Safe; $95.5\% \pm 2.0\%$ SYBR Safe labeled cells were labeled by EUB338. Scale bars, 50 μm .

Figure R1. Identification of root-colonized bacteria by DAPI staining. **A.** Representative images of bacteria-inoculated root samples labeled by the FISH probe EUB338 and DAPI. **B.** We found that $\sim 90\%$ EUB338 labeled cells were labeled by DAPI, which validated the use of FISH in detection of root-colonized bacterial cells. We noted that $\sim 40\%$ DAPI labeled regions were not labeled by EUB338 FISH probe. Our results are consistent with previous findings that the signal-to-noise ratio of SYBR is superior to DAPI (Ref 47).

Line 176: Root-colonized bacterial cells were detected by the universal FISH probe EUB338, and further validated by nucleic acid staining with SYBR Safe (Ref 42, 47) (Supplementary Fig. 9). We imaged roots colonized by two different bacterial communities (SynCom13, $n=3$; SynCom22, $n=3$). In total, we found that $97.8\% \pm 1.5\%$ EUB338 labeled cells were labeled by SYBR, and $95.5\% \pm 2.0\%$ SYBR labeled cells were labeled by EUB338 (Supplementary Fig. 9B). Also, there was no signal for negative control FISH probe NON338 on bacteria-colonized roots. Thus we demonstrate that bacteria cells colonized on rhizoplane can be correctly identified by FISH (Ref 7, 42).

2) Regarding 3 root comparisons, the spatial analysis of microbial neighborhoods are missing. For instance, authors should zoom into regions and quantify single-cells that are located near each other. Spatial distributions across tip length can also be done other techniques even using root cutting + PCR, but seqFISH can map out cellular interactions in local regions of the data. Right now, the comparison data is rather weak. What do we learn from root variations, anything we did not know before?

Response: Following the reviewer's suggestion, we have expanded spatial analysis at single-cell resolution and discussion on biological findings in the context of literature. Using the linear dipole algorithm, we found non-random spatial correlation of bacterial cells with distance up to $\sim 30 \mu\text{m}$ (Figure 5D), indicating that root-colonized bacterial cells form clusters. Indeed, we identified clusters of bacterial cells ranging from tens to hundreds of square micrometers (Figure 5E). We further zoomed into the clusters and found that some clusters consisted of bacterial cells from multiple species (Figure 5F). Using the methods adapted from Valm, A.M. et al (Ref 24), we performed spatial neighborhood analysis to identify intertaxon associations

at microscale (Figure 5G). Compared to the contact frequency between randomly distributed cells, there were 15 significant spatial associations among 9 species colonized on roots (Fig. S13). For example, we found non-random cross-correlation and contact frequency between AD1 (*Acidovorax* sp.) and AG1 (*Agrobacterium* sp.). Visual inspections of zoomed-in images (Figure 5F) also confirmed the clustering of AD1 and AG1 cells. Moreover, AH1 (*Achromobacter* sp.) and AC (*Acinetobacter* sp.) cells, which were not abundant in the community, appeared frequently in clusters and had significant associations with AD1 and AG1 cells.

The clustering of bacterial cells colonized on plant surface has been previously reported (Ref 4, 42, 51). Clusters can be formed via the growth of microcolonies upon successful colonization (Tzipilevich, E., Cell Host & Microbe, 2021); preferential attachment may also play a role, as previous studies have shown co-localization of immigrant and resident bacterial cells (Steinberg, S. et.al., The ISME Journal, 2021). Formation of clusters on plant surface is critical for bacterial fitness under environmental stress (Danhorn, T. and Fuqua, C., Annu. Rev. Microbiol., 2007). For example, it has been proposed that phyllosphere bacteria form clusters to deal with desiccation stress (J.-M. Monier and S. E. Lindow, PNAS, 2003.). Our observation of multiple bacterial species in clusters lends some support to the hypothesis of preferential attachment (Grinberg, M. et.al., PLoS Comput Biol., 2019; Steinberg, S., 2021, The ISME Journal), and future investigations along this line will deepen our understanding of microbiome assembly in rhizosphere. Moreover, the microscale intertaxon associations revealed by the spatial neighborhood analysis (Ref 24) may be indicative of short-range interactions (Dal Co, A. et.al., Nat Ecol Evol., 2020) (e.g. quorum sensing, metabolic cross feeding) and will guide mechanistic studies on the ecology of complex microbial communities.

Figure 5. E. The distribution of cluster area. F. Representative images of clusters. Scale bars, 10 μm . The clusters are surrounded by white lines and cells outside clusters are shown with a dimmer color. G. Intertaxon spatial associations observed for 12-species bacterial communities colonized on roots. Each edge shows non-random contact between two species (Supplementary Fig. 13, Methods). The width of edges is proportional to the fold increase in contact frequency

compared to randomly distributed cells. The size of nodes is proportional to the relative abundance (log transformed) of each species.

Fig. S13. Model cells to determine the frequency of random associations. **A.** Model cell images. Bacterial cells were modelled as ellipses; the lengths of major and minor axes were determined by images (Figure 2C). **B.** Simulation of contact frequency of randomly distributed cells. Contacts in the representative simulated image were indicated by white arrows. The density of each species was determined by the measured density on root samples. **C.** The contact frequency between different species in simulations (dark gray) and on roots (light gray). Statistically significant associations ($P < 0.05$) were shown. Error bars indicate the standard deviation of replicates ($n=3$) or independent simulations ($n=10$).

Line 199: We identified clusters of bacterial cells ranging from tens to hundreds of square micrometers (Fig. 5E). We further zoomed into the clusters and found that some clusters consisted of bacterial cells from multiple species (Fig. 5F). Furthermore, we performed spatial neighborhood analysis to identify non-random intertaxon associations (Ref 24) (Fig. 5G, Methods). Compared to the contact frequency between randomly distributed cells, there were 15 significant spatial associations among 9 species colonized on roots (Supplementary Fig. 13). For example, we found non-random cross-correlation and contact frequency between AD1 (*Acidovorax* sp.) and AG1 (*Agrobacterium* sp.). Visual inspections of zoomed-in images (Fig. 5F) also showed clustering of AD1 and AG1 cells. Moreover, AH1 (*Achromobacter* sp.) and AC (*Acinetobacter* sp.) cells appeared frequently in clusters and had significant associations with AD1 and AG1 cells.

Line 286: The clustering of bacterial cells colonized on plant surface has been previously reported (Ref 4, 42, 51). Clusters may be formed via the growth of microcolonies upon successful colonization (Ref 52); preferential attachment may also play a role, as previous studies have shown co-localization of immigrant and resident bacterial cells (Ref 53). Formation of clusters on plant surface may be critical for bacterial fitness under environmental stress (Ref 54). For example, it has been proposed that phyllosphere bacteria form clusters to deal with desiccation stress (Ref 51). Our observation of multiple bacterial species in clusters may lend support to the mechanism of preferential attachment (Ref 53, 55), and future investigations along this line will deepen our understanding of microbiome assembly in rhizosphere. Moreover, the microscale intertaxon associations revealed by the spatial neighborhood analysis (Ref 24) may be indicative of short-range interactions (Ref 56) (e.g. quorum sensing, metabolic cross feeding) and will guide mechanistic studies on the ecology of complex microbial communities.

Line 493: Methods: Spatial neighborhood analysis

For each pair of bacteria species, contact events between cells from each species were counted. The contact frequency was calculated as the number of contact events divided by the area of the root. In simulations, cells were randomly placed and the density of each species was determined by the measured density on root samples. Model cells were simulated as an ellipse, and the single-cell images of each bacteria species were analyzed to determine the length of major and minor axes of ellipses.

3) Please remove the novel statement from the abstract. The updated manuscript made it clear that this technique is in fact seqFISH + HD from MERFISH, no novelty here. Also multiplexing was also indicated to be similar with CLASI-FISH and HiPR-FISH and par-seqFISH. Authors should only focus on the biological findings from the experiments as an application of existing FISH methods that are already used in this platform.

Response: Following the reviewer’s suggestion, we have revised the abstract to “SEER-FISH provides a useful method for profiling the spatial ecology of complex microbial communities *in situ*”. Our method is indeed inspired by both seqFISH and MERFISH, which were developed in the context of mammalian cells for single cell transcriptomics. We have done extensive methods development to achieve accurate spatial mapping of complex microbial communities. In parallel with par-seqFISH (Dar, D. et al, Science, 2021), we are among the first to show that sequential FISH can be achieved for microbes (>25 rounds, Figure S1C). In the revised manuscript, we have included additional experimental data to illustrate the strength and novelty of our method.

1) Multiplexing: To demonstrate that the multiplexity of SEER-FISH can be extended to label more targets in complex microbial communities, we imaged a synthetic community composed of 130 bacteria strains colonized on *Arabidopsis thaliana* roots. We optimized the protocol to improve the hybridization efficiency of FISH probes and the signal-to-noise ratio for plant samples. Briefly, samples were pretreated with lysozyme to increase the permeability of cells and followed by incubation with high-concentration formamide for thorough denaturation of rRNA; in addition, the concentration of target probes was reduced to avoid background residual.

As a result, we successfully labelled 90 microbial target taxa on roots (Figure S14), exceeding the number of taxa previously shown by HiPR-FISH and CLASI-FISH.

Fig. S14. Spatial profiling of a 130-strain microbial community colonized on *Arabidopsis* root. **A.** Representative images of 130 strains colonized on *Arabidopsis* root. The 130 strains were grouped into 90 targets based on the similarity of 16S rRNA sequences. Information of strains and probe sequences are provided in Supplementary Table 3. Around 40,000 bacterial cells were identified in 5 FOVs along the root. Numbers below the image indicates the distance to the root tip. Scale bar, 200 μ m. **B.** The relative abundance of 90 target taxa detected in 5 FOVs. The numbers on the bottom indicate the number of FOVs where a given target taxa was not detected.

Line 274: With the optimizations above, we applied SEER-FISH to image a complex community composed of 130 strains colonized on *Arabidopsis* roots (Supplementary Fig. 14). The 130 strains were grouped into 90 target taxa based on the similarity of 16S rRNA sequences. In the proof-of-concept experiment, we found that all 90 taxa were successfully identified, with relative abundance ranging from 0.1% to 10%.

Line 421: In particular, for imaging 130-strains colonized *Arabidopsis* root, fixed root in FCS flow chamber were pretreated for the improvement of probe hybridization efficiency before SEER-FISH imaging. To increase the permeability of cells, sample was flowed by lysozyme solution (10 mg/mL) for 2 min and with 3-min incubation at 37°C. After PBS washing, samples were flowed by 85% formamide for 2 min and with 8-min incubation at 46°C for thorough denaturation of rRNA. These sample pretreatments greatly improved the probe hybridization efficiency and the concentration of each target probe is decreased to 15 nM accordingly, which avoids background residual caused by the high total concentration of all probes in each round. Multi-round FISH imaging was carried out as described above. All bacteria were labeled with the universal bacterial probe EUB338 during the first round of imaging. The concentration of

each target probe in each round is 250 nM for 12-strains inoculated roots and or 15 nM for 130-strains inoculated roots. The duration of probe hybridization was extended to 8 min and 15 min for 12-strains or 130-strains inoculated roots, respectively.

2) Error-robust encoding: Hamming distance, originally formulated by Richard W. Hamming for error detection and correction, has been extensively used in information theory and various applications (e.g. telecommunication). Both MERFISH and SEER-FISH imposed minimal Hamming distance on barcodes to tolerate and correct detection errors. However, the design principle of error-robust codes was based on the understanding of error rates in different systems. For mRNA labeling, MERFISH used a sparse binary HD code to minimize dropout errors (1→0), which they found to be more common than non-specific calling. In contrast, for rRNA labeling, errors are predominated by non-specific calling; thus we designed a multi-color HD code for SEER-FISH (for more explanations, please see our response on page 12-13).

For a given set of FISH probes, codebooks can be optimized to account for non-specificity of probes and improve precision and recall. In the revised manuscript, we provided an illustration of codebook optimization and its practical use in design of barcodes. We used the measured probe specificity (Figure S6) to predict the F1 score of codebooks for the 12-species synthetic community SynCom12. The codebook that we used for imaging SynCom12 had high predicted F1 score (predicted F1 score=0.92); we found that the fraction of unidentified cells was 0.11 ± 0.01 in experiments, consistent with simulation results (Figure S16 A-B). In comparison, we randomly picked a codebook with low predicted F1 score (predicted F1 score=0.12) and performed an independent imaging experiment on SynCom 12; we found that the fraction of unidentified cells increased to 0.19 ± 0.01 , and the recall rate of PD1 species was substantially lower (Figure S16 A-C). To label a community with 12 targets, there are more than 6.7×10^{20} sets of R8HD4 (S=12) possible codebooks, which cannot be enumerated for evaluation. Thus, we developed a genetic algorithm to optimize the error-robust codebook of SEER-FISH to achieve high F1 score (Figure S16 D-E, Methods). Given the information of non-specificity of FISH probes (experimentally measured or predicted), this computational approach can guide the design of error-robust encoding schemes.

Fig. S16. Codebook optimization. **A.** Two codebooks with low and high predicted F1 score. **B.** The fraction of unidentified cells in experiments and simulations, for 2 set of codebooks. Error-bars are the standard deviation of FOVs ($n > 3$) in experiments or independent simulations ($n = 10$). **C.** The recall rate of PD1. **D.** The distribution of predicted F1 score of 10000 randomly generated R8HD4 ($S = 12$) codebooks. To simulate the F1 score, the fluorescence intensity P_{pi} (probe p hybridized to species i) is set by experimental measurement (Figure S6, Methods). The codebook used in Figure 2 has a high predicted F1 score (green arrow). **E.** The optimization of F1 score via genetic algorithm. The gray dots indicate the F1 score of random generated codebooks at Round 1. The red dots indicate the 15 codebooks with the highest F1 score at each round, and the red line indicates the optimization of the best codebook. The orange dots indicate the 15 best codebooks from the previous round. The green dots indicate the codebooks evolved from the previous round.

Line 254: For a given set of FISH probes, codebooks can be optimized to account for non-specificity of probes and improve precision and recall. Here we provide an illustration of codebook optimization and its practical use in design of barcodes. We used the measured probe specificity (Supplementary Fig. 6) to predict the F1 score of codebooks for the 12-species synthetic community SynCom12. The codebook that we used for imaging SynCom12 had high predicted F1 score (predicted F1 score=0.92); we found that the fraction of unidentified cells was 0.11 ± 0.01 in experiments, consistent with simulation results (Supplementary Fig. 6). In comparison, we randomly picked a codebook with low predicted F1 score (predicted F1 score=0.12) and performed an independent imaging experiment on SynCom 12; we found that the fraction of unidentified cells increased to 0.19 ± 0.01 , and the recall rate of PD1 species was substantially lower (Supplementary Fig. 16 A-C). To label a community with 12 targets, there are more than 6.7×10^{20} sets of R8HD4 ($S = 12$) possible codebooks, which cannot be

enumerated for evaluation. Thus, we developed a genetic algorithm to optimize the error-robust codebook of SEER-FISH to achieve high F1 score (Supplementary Fig. 16 D-E, Methods). Given the information of non-specificity of FISH probes (experimentally measured or predicted), this computational approach can guide the design of error-robust encoding schemes.

Line 464: To predict the F1 score with measured probe specificity, the log-transformed fluorescence intensity ($\log(P_{pi})$) is drawn from a normal distribution

$$f(x) = \frac{1}{\sigma\sqrt{2\pi}} e^{-\frac{1}{2}\left(\frac{x-\mu}{\sigma}\right)^2}$$

where μ and σ is set as the mean and standard deviation of the log-transformed intensity when probe (with corresponding fluorophores) p is hybridized to species i .

Line 470: Methods: Codebook optimization

We used the genetic algorithm to optimize codebooks to achieve high F1 score. Briefly, for the first round, 200 sets of codebooks were randomly generated and 15 codebooks with highest F1 score were selected. For each codebook selected from the previous round, 12 mutated codebooks were generated by random shuffling of the barcode for one of the 12 species. The 15 selected codebooks from the previous round, 180 mutated codebooks, and 5 randomly generated codebooks, were evaluated together to select for 15 codebooks with highest F1 score for the next round. This process can be repeated until the codebook with desired F1 score has been found.

4) For multiplexing, it talks about 10^5 multiplexing, but this paper indicated the use of only 12 to 30 probes (supplementary table 2). Authors should remove highly multiplexed claims from this manuscript without demonstrating that thousands multiplexing is feasible because they are unsure about density of microbes.

Response: We have removed the statement on the 10^5 multiplexity from the manuscript. In the response above, we demonstrate experimentally that multiplexing is feasible with 90 probes.

5) HD correction in MERFISH is said to be related to drop outs but not non specific binding. In fact, MERFISH or seqFISH error correction (<https://www.ncbi.nlm.nih.gov/pmc/articles/PMC6544023/>) both captures drop out or nonspecific calling. authors should remove this statement as it may not be correct. Any barcode does not correspond to the original assignment will be removed, making the correction regardless of false positives or negatives.

Response: We fully agree with the reviewer that error correction in MERFISH and seqFISH can capture both dropout and nonspecific calling. What we pointed out was that non-specific calling is less common than dropout errors in mRNA labeling. In seqFISH (Shah, S. et al, Neuron, 2016), the authors also pointed out that “our simple error correction accounts of the most common error, dropped signal”. In MERFISH (Chen, K.H. et al, Science, 2015), the authors estimated that $1 \rightarrow 0$ error rates were 10% on average, while $0 \rightarrow 1$ error rates were 4% on average. Thus, in MERFISH, modified HD4 codes (with only four “1” bits in 16-bit barcode) were used to minimize dropout errors ($1 \rightarrow 0$).

In contrast, for rRNA labeling in microbial communities, errors are predominated by non-specific calling; dropout errors $1 \rightarrow 0$ are negligible due to the high abundance of rRNA. While non-specific binding cannot be completely avoided for rRNA FISH probes (the target region for probe design is limited), the fluorescence intensity of the specific probe is on average much higher than nonspecific probes (Figure 2F). Therefore, in our image analysis, the color code of each cell in each round was determined by the brightest fluorescence channel in the corresponding round (Methods). Because $F \rightarrow F'$ errors can be better avoided than $0 \rightarrow F'$ errors, for the barcode design of SEER-FISH, we chose to exclude the non-fluorescent code in the codebook (i.e., color code=0) to minimize detection errors caused by non-specific labeling (color code $0 \rightarrow F'$).

Also, we showed that, if we simply remove all barcodes that did not correspond to the original assignment (i.e. only keeping “perfect match”), the recall rate would be substantially lower (Figure 2D, Figure 3F); thus incorporating 1/2/3-bit error correction in barcode identification is very important for the performance of microbial identification by SEER-FISH. We have revised the manuscript to clarify all the points above.

Line 244: In mRNA labeling, non-specific calling is less common than dropout errors (Ref 32, 50). Thus, in MERFISH, modified HD4 codes (with only four “1” bits in 16-bit barcode) were used to minimize dropout errors ($1 \rightarrow 0$). In contrast, detection errors in bacterial rRNA FISH are mainly caused by non-specific (i.e., off-target) labeling of phylogenetically related rRNA sequences; dropout errors $1 \rightarrow 0$ are negligible due to the high abundance of rRNA. While non-specific binding cannot be completely avoided for rRNA FISH probes (the target region for probe design is limited), the fluorescence intensity of the specific probe is on average much higher than nonspecific probes (Fig. 2F). Therefore, in our image analysis, the color code of each cell in each round was determined by the brightest fluorescence channel in the corresponding round (Methods). Because $F \rightarrow F'$ errors can be better avoided than $0 \rightarrow F'$ errors, we chose to exclude the non-fluorescent code in the codebook (i.e., color code=0) to minimize detection errors caused by non-specific labeling (color code $0 \rightarrow F'$).

After the revision, it has become apparent that the novelty of barcode correction is no longer relevant here. The biological findings also remain weak because spatial neighborhood analysis or any functional microbial mapping have not been included that can show us something previously unknown. Thus, the revision dampened the enthusiasm towards this manuscript due to clarification of the technique to be a seqFISH only and the findings did not yield any biological novel insights. Why would you observe some bacteria in a specific location near another bacteria or why are they located in different length ranges. Also, the revision seems to be rushed with only $n=3$ roots while the biology of roots can rather be complex and biological significance may take more samples. Validation experiments have not even been quantified or statistically measured. Multiplexing is only 30 species that can also be measured by CLASI-FISH25 and HiPR-FISH and 10^5 multiplexing is unsupported because authors do not have even a good 16S classifier for these many species. It remains as a hype in this current version.

Response: We thank the reviewer again for helping us improve the manuscript. Here we

provide a brief summary of our revisions, which have been detailed in the responses above.

1) Error-robust encoding: We imposed minimal Hamming distance among barcodes to tolerate and correct detection errors, a strategy that has been widely studied in coding theory and beautifully illustrated by MERFISH. Based on our experiments and understanding on the non-specificity of rRNA FISH probes in microbial communities, the multi-color codebook of SEER-FISH was thoughtfully designed to minimize detection errors in multi-round rRNA FISH imaging. The design of our codebooks has been successfully validated in different microbial communities; our error correction strategy in barcode identification was important for achieving high recall rate. Furthermore, we provided a computational approach to optimize codebooks for higher precision and recall (Figure S16).

2) Spatial analysis: We have performed spatial neighborhood analysis to identify significant associations among root-colonized bacterial species (Figure 5 and S13). Combined with the linear dipole analysis on non-random clustering and zoomed-in visual inspections at single cell level, our spatial analysis showed that root-colonized bacteria formed clusters composed of multiple species. Our results indicate that ecological interactions are important for microbiome assembly in host-associated environments, and that associations identified by spatial profiling can guide mechanistic studies of interspecies interactions.

3) Validation experiments and sample size: We have quantified the overlapping FISH signal with DNA dye SYBR, which was highly consistent across multiple roots (Figure S9). Our results demonstrate that bacteria cells colonized on rhizoplane can be correctly identified. Moreover, among the root samples shown in Figure 4/5, we found highly similar patterns in species composition (Pearson's correlation > 0.97) and non-random spatial clustering (auto-correlation > 0 within distance ~30 μm). Thus, the number of samples in our validation experiments and spatial profiling was sufficient to reveal the similarities across roots. While the imaging experiments in this study were not designed to investigate root-to-root variations, we agree that such variations exist (e.g. caused by stochasticity in community assembly) and should be taken carefully if comparing multiple experimental groups (e.g. treatment vs control).

4) Multiplexing: We have removed the 10^5 multiplexing claim from our manuscript. With optimization of sample pretreatment and probe concentration, we demonstrated the feasibility of spatially profiling 90 microbial taxa on plant samples (Figure S14). We understand that there is no need to spatially resolve 10^5 taxa, yet as we pointed out in discussion: "the complexity of meta-transcriptomes (> 10^7 genes in human gut microbiomes) require an increase in multiplexing, which can be achieved by sequential labeling." par-seqFISH has wonderfully demonstrated that it is feasible to sequentially label bacterial mRNA to profile single-cell transcriptome in a particular bacterial species. In future studies, our method can be combined with par-seqFISH to spatially profile both taxonomy and transcriptome in complex communities.

Reviewers' Comments:

Reviewer #2:

Remarks to the Author:

With this revision the authors have satisfactorily addressed my previous concerns. They have also added a proof of concept experiment using 90 probes to image 130 bacterial strains colonizing an Arabidopsis root (Figure S14 and associated text). To achieve successful FISH with this high multiplexity, the authors adjusted the hybridization protocol, adding pre-treatment steps, reducing probe concentration, and increasing hybridization time.

This added experiment in principle increases the impact of this manuscript, in that it demonstrates the application of the method to imaging and distinguishing, in the context of the root, 90 taxa isolated from roots. The authors report that "all 90 taxa were successfully identified" in a total of 5 fields of view from one root. However, basic controls for this experiment are lacking. The imaging, analysis, and error correction pipeline resulted in successfully assigning each taxon identity code to at least some cells in the images. In order to assess the accuracy of the assignment, pure cultures or mixtures of pure cultures in known proportions should be analyzed to determine the error rate of the analysis. This was done in Figure S5 for the 12-taxon experiment; a similar validation should ideally be carried out for each taxon to be targeted in this 90-taxon, 130-strain experiment. At the very least, sample mixtures containing only a subset of the taxa should be imaged and analyzed to measure the accuracy of identification, particularly as the 90-taxon experiment was carried out using pre-treatment with lysozyme and high-concentration formamide followed by lower probe concentration and longer hybridization times than the previous experiments.

I would appreciate if the authors could include in the manuscript some comment about why it was necessary to increase the hybridization time for the more complex probe sets. The authors should also adjust table S1 to reflect the increased hybridization time.

On the subject of controls, it is difficult to understand what images are shown in Figure S6A. Each column represents images from one bacterial strain. What does each row represent? The rows are generically labeled "Probe" and the caption describes hybridization with 12 probes x 3 fluorophores x 12 rounds of FISH imaging. Which image is shown?

The authors also add analysis and discussion of the clustering of bacterial cells that they detect on Arabidopsis roots. While the finding of clustering is not particularly surprising, the analysis and discussion adds biological relevance to the manuscript.

In general the text is well-written and easy to understand. The new text contains some small grammatical or editing errors. Please remove the duplicated text lines 472-474 "Multi-round FISH imaging was carried out as previously described. All bacteria were labeled with the universal bacterial probe EUB338 during the first round of imaging." Most other grammar errors involve articles (a, an, the) and are very minor, e.g., line 90 "bacteria cells colonized on rhizoplane" should read "bacterial cells colonizing the rhizoplane" and line 522 should read "We used a genetic algorithm to optimize codebooks to achieve a high F1 score." Lines 214-215 "zoomed into the clusters" is informal language and could be more formally phrased "imaged the clusters at higher magnification"; line 220 "zoomed-in images" could be "high-magnification images".

Reviewer #3:

Remarks to the Author:

The authors have clarified their manuscript and provided more accurate descriptions of the current implementation of seqFISH and MERFISH in microbial mapping. Mostly the answers are convincing, but the Fig 5 is relatively pre-mature. After all barcoding and correction, only small regions of 3 tiny roots have been analyzed for the spatial neighborhood of 12 bacterial species. I would suggest that authors continue analyzing more plant roots and make biological conclusions with statistical power. For instance, in medical applications, this demonstration would be considered rather weak for biological findings. Thus, this manuscript needs to establish the sample power and biological depth of those bacterial species. Power statistics on sample size and the significance tests should

be provided on biological insights obtained from the plant roots. Even a perturbation on spatial re-organization of bacterial communities in the plant roots would be useful to provide a convincing result.

REVIEWER COMMENTS

Reviewer #2 (Remarks to the Author):

With this revision the authors have satisfactorily addressed my previous concerns. They have also added a proof of concept experiment using 90 probes to image 130 bacterial strains colonizing an Arabidopsis root (Figure S14 and associated text). To achieve successful FISH with this high multiplexity, the authors adjusted the hybridization protocol, adding pre-treatment steps, reducing probe concentration, and increasing hybridization time.

This added experiment in principle increases the impact of this manuscript, in that it demonstrates the application of the method to imaging and distinguishing, in the context of the root, 90 taxa isolated from roots. The authors report that "all 90 taxa were successfully identified" in a total of 5 fields of view from one root. However, basic controls for this experiment are lacking. The imaging, analysis, and error correction pipeline resulted in successfully assigning each taxon identity code to at least some cells in the images. In order to assess the accuracy of the assignment, pure cultures or mixtures of pure cultures in known proportions should be analyzed to determine the error rate of the analysis. This was done in Figure S5 for the 12-taxon experiment; a similar validation should ideally be carried out for each taxon to be targeted in this 90-taxon, 130-strain experiment. At the very least, sample mixtures containing only a subset of the taxa should be imaged and analyzed to measure the accuracy of identification, particularly as the 90-taxon experiment was carried out using pre-treatment with lysozyme and high-concentration formamide followed by lower probe concentration and longer hybridization times than the previous experiments.

Response: We thank the reviewer again for helping us improve the manuscript. Following the reviewer's suggestion, we used pure cultures of 90 taxa to evaluate the performance of taxonomic identification by SEER-FISH in highly complex communities. Each strain was separately coated onto a coverslip, then hybridized with probes according to the codebook and imaged for nine sequential rounds. The sample pre-treatment, probe concentration, hybridization time, and R9HD4 codebook in 90-taxon validation were the same as used in 130-strain *in vivo* imaging. Bacterial cells were identified by decoding their barcodes and compared with the ground truth. We found excellent precision (median=0.87) and recall (median=0.78) for most taxa (Figure S14A, Figure R1). Poor identification of some taxa can be due to strong non-specificity of probes from their closely-related taxa, or inefficient hybridization of their specific probes (see examples in Figure R2). We note that further improvement in probe design would be helpful in the application of multiplexed FISH methods to profile highly complex communities. We have revised the manuscript and Fig S14 accordingly.

Line 311: Similar to the validation experiments that we performed on 12 taxa (Fig. 2 and Supplementary Fig. 5), we used pure cultures of 90 taxa to evaluate the performance of taxonomic identification by SEER-FISH in highly complex communities (Supplementary Fig. 14A). We found excellent precision (median=0.87) and recall (median=0.78) for most taxa. As a proof-of-concept experiment, we used these 90 FISH probes to image a synthetic community of 130 strains colonized on Arabidopsis root (Supplementary Fig. 14B) and correctly identified ~65% bacterial cells. Further improvement in probe design would be helpful in the application of multiplexed FISH methods to profile highly complex communities.

Figure S14. Spatial profiling of a 130-strain microbial community colonized on *Arabidopsis* root. (A) The 130 strains were grouped into 90 taxa based on the similarity of 16S rRNA sequences. Information of strains and probe sequences is provided in Supplementary Table 3. We used pure cultures of 90 taxa to evaluate the performance of taxonomic identification by SEER-FISH. Each strain was separately coated onto a coverslip, then hybridized with probes according to the codebook and imaged for nine sequential rounds. The sample pre-treatment, probe concentration, hybridization time, and R9HD4 codebook in 90-taxon validation were the same as used in 130-strain in vivo imaging. Bacterial cells were identified by decoding their barcodes and compared with ground truth. The median and interquartile range of recall rate, precision, and F1 score are 0.78(0.51-0.92), 0.87(0.72-0.94), and 0.77(0.59-0.90), respectively. (B) Imaging of a 130-strain synthetic community colonizing on *Arabidopsis* root. Around 40,000 bacterial cells were identified in 82 FOVs along the root. Numbers below the image indicate the distance to the root tip. Scale bar, 200 μm. (C) The ratio of correctly identified cells on root was ~65%.

Figure R1. Evaluation of taxonomic identification by SEER-FISH using pure cultures of 90 taxa. (A) Recall ratio of 90 taxa measured by SEER-FISH with the 9-bit barcodes. For each taxon, cells correctly identified are true positives (Green); cells incorrectly identified as the other 89 taxon are marked as misidentified (Orange); cells that cannot be classified to any of the 90 taxa are marked as unidentified (Gray). (B) Precision of 90 taxa measured by SEER-FISH with the 9-bit barcodes. Cells of the other 89 taxa incorrectly identified as the corresponding taxon are false positives (Red). Ratios are normalized by the identified cell number of each taxon. The ranking is sorted by recall (panel A) or precision (panel B), respectively.

Figure R2. Barcodes and multi-round imaging of representative taxa. (A) Taxa with high F1 Score. (B) Taxa with Low F1 Score.

I would appreciate if the authors could include in the manuscript some comment about why it was necessary to increase the hybridization time for the more complex probe sets. The authors should also adjust table S1 to reflect the increased hybridization time.

Response: To image highly complex communities with a large number of probes, we found that the background residual caused by the high concentration of probes would hamper the identification of bacterial cells. Therefore, we decreased the concentration of each probe and also added sample pre-treatment by lysozyme to increase the permeability of cells. To account for potential effects of reduced probe concentration (15 nM for each probe) on hybridization efficiency, we increased the hybridization time. Following the reviewer's suggestion, we have included comments on the increased hybridization time in the Methods section. Table S1 and Figure S1A legends were also revised to reflect the increased hybridization time.

Line 472: The concentration of each target probe in each round was 250 nM for SynCom12 root samples, and reduced to 15 nM for SynCom30.2 and SynCom130 root samples. The duration of probe hybridization was 8 min for SynCom12 root samples, and extended to 15 min for SynCom 30.2 and SynCom130 samples. We extended the hybridization time to account for potential effects of reduced probe concentration on hybridization efficiency.

Figure S1A: sample is incubated for 3-15 min at 46°C according to the type of sample and the probe concentration used.

On the subject of controls, it is difficult to understand what images are shown in Figure S6A. Each column represents images from one bacterial strain. What does each row represent? The rows are generically labeled "Probe" and the caption describes hybridization with 12 probes x 3 fluorophores x 12 rounds of FISH imaging. Which image is shown?

Response: In Figure S6A, each row represents the specific probe designed for each strain,

ranked in the same order as columns. The images shown were based on probes labeled with Cy5 fluorophore. We have revised the Figure S6 legends to clarify the points above.

Fig. S6. Probe specificity analysis. (A) Fluorescent images of 12 pure bacterial cultures when hybridized with 12 candidate probes. 12 candidate probes were chosen to hybridize with pure cultures of target and non-target bacteria according to the codebook in Supplementary Table 5 (“codebook used for probe specificity analysis”). Each row represents the specific probe designed for each strain, ranked in the same order as columns. To check the consistency of fluorescence signals between different fluorophores, 12 specific probes with three different conjugations (FAM, Cy3, Cy5) were all examined during 12 rounds of FISH imaging (see Supplementary Table 5). The images of three fluorescence channels and phase-contrast images were collected for all species. The images shown were based on probes labeled with Cy5 fluorophore. The fluorescence intensity of displayed images was normalized by the highest intensity in each column.

The authors also add analysis and discussion of the clustering of bacterial cells that they detect on *Arabidopsis* roots. While the finding of clustering is not particularly surprising, the analysis and discussion adds biological relevance to the manuscript

Response: We thank the reviewer for acknowledging the insights provided by the spatial analysis. We have included discussion on the clustering of bacterial cells and its biological relevance (Line 322-330).

In general the text is well-written and easy to understand. The new text contains some small grammatical or editing errors. Please remove the duplicated text lines 472-474 "Multi-round FISH imaging was carried out as previously described. All bacteria were labeled with the universal bacterial probe EUB338 during the first round of imaging." Most other grammar errors involve articles (a, an, the) and are very minor, e.g., line 90 "bacteria cells colonized on rhizoplane" should read "bacterial cells colonizing the rhizoplane" and line 522 should read "We used a genetic algorithm to optimize codebooks to achieve a high F1 score." Lines 214-215 "zoomed into the clusters" is informal language and could be more formally phrased "imaged the clusters at higher magnification"; line 220 "zoomed-in images" could be "high-magnification images".

Response: We thank the reviewer for detailed suggestions. We have revised the texts accordingly.

Reviewer #3 (Remarks to the Author):

The authors have clarified their manuscript and provided more accurate descriptions of the current implementation of seqFISH and MERFISH in microbial mapping. Mostly the answers are convincing, but the Fig 5 is relatively pre-mature. After all barcoding and correction, only small regions of 3 tiny roots have been analyzed for the spatial neighborhood of 12 bacterial species. I would suggest that authors continue analyzing more plant roots and make biological conclusions with statistical power. For instance, in medical applications, this demonstration would be considered rather weak for biological findings. Thus, this manuscript needs to establish the sample power and biological depth of those bacterial species. Power statistics on sample size and the significance tests should be provided on biological insights obtained from the plant roots. Even a perturbation on spatial re-organization of bacterial communities in the plant roots would be useful to provide a convincing result.

Response: We thank the reviewer for acknowledging the value of our study and helping us improve the manuscript. Following the reviewer's suggestion, we conducted additional experiments to perturb the spatial organization of bacterial communities by plant metabolites camalexin and fraxetin (**Figure 6, Supplementary Figure 17-19**). Camalexin is one of the alkaloid phytoalexin produced and secreted by *Arabidopsis* in response to pathogen invasion and has been reported to affect plant-microbe interactions. Fraxetin belongs to the group of coumarin and is typically synthesized and secreted by *Arabidopsis* roots under iron deficiency. Root-secreted metabolites have been found to regulate the composition of rhizosphere microbiome, but their effects on the spatial organization of microbiome remain largely unknown.

We increased the sample size of imaged roots (n=10) for each experimental group (control, camalexin-treated and fraxetin-treated) and provided significance tests in spatial analysis. The imaged regions covered four developmental zones of the root tip, including the root cap, the meristematic zone, the elongation zone and the differentiation zone. We found that the sample size used for imaging was sufficient to account for root-to-root variations (Figure 6B) and to reveal the perturbations by plant metabolites (Figure 6C). As described in details below, our experiments demonstrated that plant metabolites significantly altered the community composition, spatial distribution along root and intertaxon spatial associations at micron-scale. In summary, we have expanded the spatial analysis of root-colonized microbial communities substantially in the revised the manuscript.

Line 213 (Results):**Perturbation on the spatial organization of root-colonized microbial communities by plant metabolites**

Plant metabolites have been shown to modulate the composition and function of plant-associated microbiome. Camalexin is one of the alkaloid phytoalexin produced and secreted by *Arabidopsis* in response to pathogen invasion and has been reported to affect plant-microbe interactions. Fraxetin belongs to the group of coumarin and is typically synthesized and secreted by *Arabidopsis* roots under iron deficiency. Fraxetin is also recognized for its antimicrobial function.

Here we applied SEER-FISH to study the effects of camalexin and fraxetin on the spatial organization of root-colonized microbial communities (**Supplementary Fig. 17A**). The 30-

strain microbial community (SynCom30.2) spanned the phylogenetic diversity of *Arabidopsis* rhizosphere microbiome and included members that were previously shown to respond to plant metabolites. Our *in vitro* growth experiments also confirmed selective growth modulation of camalexin and fraxetin on members of the community (**Supplementary Fig. 17C**). For each root, ~80 FOVs were captured (within ~4mm from the tip) (**Fig. 6A-C**). The compositional profiles given by SEER-FISH imaging (n=10 roots) were in good agreement with the profiles given by 16S amplicon sequencing (**Supplementary Fig. 17B**). For camalexin-treated and fraxetin-treated plants, root-colonized microbiota showed clear shifts in composition compared to plants in the control group (**Fig. 6C, Supplementary Fig. 17C**). For example, the abundance of *Mesorhizobium* sp. decreased under camalexin/fraxetin treatment, consistent with the observation that its growth was strongly inhibited by the plant metabolites *in vitro*.

We examined the spatial distribution of microbial colonization along the root and the perturbations imposed by plant metabolites (**Fig. 6D-E and Supplementary Fig. 18**). For example, *Sinorhizobium* was abundant in the control group and mostly colonized the region within 1mm to the tip. We found that the abundance and the spatial pattern of *Sinorhizobium* strains were significantly altered by camalexin and fraxetin (**Fig. 6D, Supplementary Fig. 18C**). In contrast, the spatial distribution of *Agrobacterium* sp. was uniform in the control group; in fraxetin-treated plants, *Agrobacterium* sp. showed preferential colonization near the maturation zone (**Fig. 6E**). These non-uniform and taxon-specific spatial patterns of root-colonized microbes indicate strong heterogeneity in root environments (e.g. region-specific exudates) as well as diverse microbial traits. Furthermore, we found that plant metabolites disrupted the spatial associations between several bacterial taxa (**Fig. 6F and Supplementary Fig. 19, Methods**). Camalexin treatment significantly increased spatial association between *Acidovorax* sp. and *Arthrobacter* sp. 2 in (**Supplementary Fig. 19A**), while fraxetin treatment altered spatial associations between *Lysobacter* sp. and *Sinorhizobium* strains (**Supplementary Fig. 19B**).

Line 322 (Discussion):

The application of SEER-FISH on plant samples has revealed the micron-scale spatial organization of root-colonized microbial communities, including clustering of multiple species and intertaxon spatial associations. The clustering of bacterial cells colonized on plant surface has been previously reported. Clusters can form via the growth of microcolonies upon successful colonization. Formation of clusters on plant surface may be critical for bacterial fitness under environmental stress. For example, it has been proposed that phyllosphere bacteria form clusters to deal with desiccation stress. Preferential attachment is another potential mechanism for the formation of clusters, as previous studies have shown co-localization of immigrant and resident bacterial cells. Our observation of multiple bacterial species in clusters may lend support to the hypothesis of preferential attachment. Furthermore, the micron-scale intertaxon spatial associations may be indicative of short-range interactions (e.g. quorum sensing, metabolic cross feeding, niche competition, contact-dependent inhibition) and will guide mechanistic studies on the ecology of complex microbial communities. Lastly, alterations in the spatial structure of microbiome during host development, stress response (biotic and abiotic) and diseases could lead to novel insights in host-microbiome interactions. Root-secreted metabolites have been found to regulate the composition of rhizosphere microbiome,

but their effects on the spatial organization of microbiome remain largely unknown. Future investigations along these lines will deepen our understanding of microbiome assembly in rhizosphere/phylosphere and its implications to plant fitness.

Line 543 (Methods):

Spatial association analysis

The spatial association between two taxa was calculated as the number of co-occurrence events inside the association range of 10 μm *in planta* compared to a randomly generated co-occurrence test. The test was done by randomly assigning the taxa in the same measured images while keeping the abundance unchanged. For each root, 100 random tests were performed, and the association was calculated as the fold change between the measured co-occurrence and the mean value of random simulated tests. If the measured co-occurrence events were too low, the corresponding measurements are dropped. The spatial association analysis was performed for each experimental group, and differential spatial association was calculated as the fold change between the treatment group and the control group. The statistical significance was determined by independent t-test and the significance threshold level was adjusted by Bonferroni correction. The number of hypotheses tested was calculated as the number of unique taxon–taxon associations detected across all roots.

Figure 6. Perturbation on the spatial organization of root-colonized microbial communities by plant metabolites. **A**) Representative images of microbial communities in the meristem and elongation zone (~200 μm from the tip, top panels) and in the differentiation zone (~1.7 mm from the tip, bottom panels). Scalebar, 50 μm . **B**) The number of imaged microbial cells and the community compositional profiles given by SEER-FISH. The number of cells imaged by SEER-FISH for a root sample in the control group, the camalexin-treated group and the fraxetin-treated group were $2.7 \pm 0.8 \times 10^4$, $3.1 \pm 1.3 \times 10^4$ and $2.5 \pm 1.2 \times 10^4$, respectively. For each experimental group, 10 roots were imaged. For each root, ~80 FOVs were captured (within ~4mm from the root tip). **C**) Principal Coordinate Analysis (PCoA) based on Bray-Curtis

dissimilarity of community compositional profiles given by imaging. Solid square indicates the compositional profile averaged over 10 root samples. **D-E)** Spatial distribution of *Sinorhizobium* sp. 2 (panel D) and *Agrobacterium* sp. (panel E) along the root. Error bars are SEMs (n=10 roots). **F)** Differential spatial association analysis on root-colonized microbial taxa between camalexin-treated (or fraxetin-treated) plants and control plants. Fold change refers to $\log_2[(\text{association frequency on camalexin-treated (or fraxetin-treated) roots}/\text{simulated random frequency in treated roots})/(\text{association frequency on control roots}/\text{simulated random frequency in control roots})]$. Gray areas indicate that the analysis is not applicable.

Figure S17. Perturbation on the community profile by camalexin and fraxetin. (A) Illustration of the experiment setup. A synthetic community of 30 strains (SynCom30.2) was used to colonize *Arabidopsis*. For treatment groups, camalexin or fraxetin was introduced into MS media (Methods). (B) The composition of the 30-strain synthetic microbial community colonized on root measured by SEER-FISH (n=10, within ~4mm from root tip) and by 16S amplicon sequencing (n=3 samples, in each sequencing sample 4 roots were pooled for DNA extraction). The Spearman correlation between the compositional profile given by SEER-FISH and the profile given by 16S amplicon sequencing (***: p<0.001) is shown for each experimental group. (C) Fold change in the abundance of 30 bacterial strains under treatment of plant metabolites *in vitro* (monoculture, measured by Optical Density) and *in planta*. Seq and FISH indicate the fold change in abundance measured by sequencing and by imaging. Gray areas: fold change is not applicable due to low abundance. *: P<0.05, unpaired Student's t-test.

Figure S18: Spatial distribution of microbial taxa along roots. (A) The spatial distribution of microbial cells identified by SEER-FISH. (B) The spatial self-correlation of root-colonized bacterial cells is analyzed by linear dipole algorithm (Methods). The solid lines indicate the correlation between bacterial cells, the shadows indicate the 95% confidence intervals (estimated by sampling $n=10$ roots). The horizontal dashed line ($g(r)=1$) refers to the expected value of random distribution. (C) Spatial distribution of 6 representative strains. (D) For each experimental group, Pearson correlation is calculated between the local cell number (each local area is 200 μm in length) and the distance to root tip. Positive (or negative) correlation means that the local abundance of the corresponding strain increases (or decreases) with the distance to root tip.

Figure S19: Spatial association analysis of root-colonized microbial communities. (A-B) Differential spatial association between camalexin-treated and control group (panel A), or between fraxetin-treated treated and control group (panel B). The horizontal dashed line indicates the significance threshold determined by Bonferroni correction (Methods). Statistically significant spatial associations are listed below each panel. **(C)** Spatial association network of each experimental group (Methods).

Reviewers' Comments:

Reviewer #2:

Remarks to the Author:

With this re-revision the authors have fully addressed my previous concerns.

The figure R1 included in the Response to Reviewers provides interesting and useful information, and I recommend that it be included in the manuscript as a supplemental figure.

In the methods for multi-round FISH imaging, lines 410 and 417, could the authors please confirm whether the concentration of EDTA in the wash buffer and the dissociation buffer was in fact 5 micromolar rather than perhaps 5 millimolar?

This manuscript represents a substantial experimental effort and I believe it makes an important contribution to the field.

Reviewer #3:

Remarks to the Author:

Authors have addressed the concerns raised by the reviewer. Biological findings from perturbation experiments by Camalexin and Fraxetin demonstrate the potential application of SEER-FISH in plant biology.

Only minor concern is Fig 6F should cover the full window width, allowing to show the taxonomy and bacterial distributions with larger fonts and more details.

REVIEWERS' COMMENTS

Reviewer #2 (Remarks to the Author):

With this re-revision the authors have fully addressed my previous concerns.

Response: We thank the reviewer again for helping us improve the manuscript.

The figure R1 included in the Response to Reviewers provides interesting and useful information, and I recommend that it be included in the manuscript as a supplemental figure.

Response: Following the reviewer's suggestion, the figure R1 has been added in Supplementary Fig. 19.

In the methods for multi-round FISH imaging, lines 410 and 417, could the authors please confirm whether the concentration of EDTA in the wash buffer and the dissociation buffer was in fact 5 micromolar rather than perhaps 5 millimolar?

Response: We thank the reviewer for pointing out the typo. We have corrected the concentration of EDTA to 5 millimolar.

This manuscript represents a substantial experimental effort and I believe it makes an important contribution to the field.

Reviewer #3 (Remarks to the Author):

Authors have addressed the concerns raised by the reviewer. Biological findings from perturbation experiments by Camalexin and Fraxetin demonstrate the potential application of SEER-FISH in plant biology.

Response: We thank the reviewer again for helping us improve the manuscript.

Only minor concern is Fig 6F should cover the full window width, allowing to show the taxonomy and bacterial distributions with larger fonts and more details.

Response: We have revised the Fig. 6f following the reviewer's suggestion.